# Integrated omics endotyping of infants with respiratory syncytial virus bronchiolitis and risk of childhood asthma

Yoshihiko Raita [1 ✉], Marcos Pérez-Losada[2,3], Robert J. Freishtat[4,5,6], Brennan Harmon[4], Jonathan M. Mansbach[7], Pedro A. Piedra[8], Zhaozhong Zhu[1], Carlos A. Camargo [1] & Kohei Hasegawa [1]

Respiratory syncytial virus (RSV) bronchiolitis is not only the leading cause of hospitalization in U.S. infants, but also a major risk factor for asthma development. While emerging evidence suggests clinical heterogeneity within RSV bronchiolitis, little is known about its biologically-distinct endotypes. Here, we integrated clinical, virus, airway microbiome (species-level), transcriptome, and metabolome data of 221 infants hospitalized with RSV bronchiolitis in a multicentre prospective cohort study. We identified four biologically- and clinically-meaningful endotypes: A) clinical$^{classic}$microbiome$^{M.\ nonliquefaciens}$inflammation$^{IFN-intermediate}$, B) clinical$^{atopic}$microbiome$^{S.\ pneumoniae/M.\ catarrhalis}$inflammation$^{IFN-high}$, C) clinical$^{severe}$microbiome$^{mixed}$inflammation$^{IFN-low}$, and D) clinical$^{non-atopic}$microbiome$^{M.catarrhalis}$inflammation$^{IL-6}$. Particularly, compared with endotype A infants, endotype B infants—who are characterized by a high proportion of IgE sensitization and rhinovirus coinfection, S. pneumoniae/M. catarrhalis codominance, and high IFN-α and -γ response—had a significantly higher risk for developing asthma (9% vs. 38%; OR, 6.00: 95%CI, 2.08–21.9; P = 0.002). Our findings provide an evidence base for the early identification of high-risk children during a critical period of airway development.

[1] Department of Emergency Medicine, Massachusetts General Hospital, Harvard Medical School, Boston, MA, USA. [2] Computational Biology Institute, Department of Biostatistics and Bioinformatics, The George Washington University, Washington, DC, USA. [3] CIBIO-InBIO, Centro de Investigação em Biodiversidade e Recursos Genéticos, Universidade do Porto, Campus Agrário de Vairão, Vairão, Portugal. [4] Center for Genetic Medicine Research, Children's National Hospital, Washington, DC, USA. [5] Division of Emergency Medicine, Children's National Hospital, Washington, DC, USA. [6] Department of Pediatrics, George Washington University School of Medicine and Health Sciences, Washington, DC, USA. [7] Department of Pediatrics, Boston Children's Hospital, Harvard Medical School, Boston, MA, USA. [8] Departments of Molecular Virology and Microbiology and Pediatrics, Baylor College of Medicine, Houston, TX, USA. ✉email: yraita1@mgh.harvard.edu

Bronchiolitis is the leading cause of hospitalization in U.S. infants, accounting for ~110,000 hospitalizations with the direct cost of $734 million annually[1]. In addition to the substantial acute morbidity, ~30% of infants hospitalized for bronchiolitis ("severe bronchiolitis") subsequently develop asthma in childhood[2]. Of the major causative pathogens, respiratory syncytial virus (RSV) infection during infancy has the largest population attributable fraction in asthma development (i.e., the most impactful risk factor)[3].

While bronchiolitis has been considered a single disease with similar mechanisms[2], emerging evidence suggests heterogeneity in clinical presentations[4] and chronic morbidities (e.g., subsequent risk of recurrent wheeze and asthma)[5,6]. Growing evidence also suggests distinct upper airway microbiome[7,8], transcriptome[9], cytokine[10,11], and metabolome[12] profiles among infants with bronchiolitis. However, these findings have been solely derived from single-level (e.g., clinical or microbiome) data. Understanding the complex interplay among the host, respiratory viruses, airway microbiome, and subsequent chronic morbidities of bronchiolitis involve several major challenges—e.g., identification of the responsible mechanisms (e.g., host–microbiome interrelations), the effect of clinical factors reflecting multi-level environmental relations, and heterogeneity of bronchiolitis itself. To our knowledge, no study has integrated clinical, virus, and multi-omics data to investigate the endotypes of RSV bronchiolitis and their longitudinal relations with chronic morbidities in childhood. Our still limited understanding of the heterogeneity of RSV bronchiolitis during infancy—an important period of lung development—has hindered efforts to develop endotype-specific RSV bronchiolitis treatment and asthma prevention strategies in this large patient population with substantial morbidity burden.

To address this knowledge gap in the literature, analyzing data from a multicentre prospective cohort, we sought to (1) identify biologically distinct RSV bronchiolitis endotypes through applying integrative network and clustering approaches to clinical, virus, nasopharyngeal airway microbiome, transcriptome, and metabolome data and to (2) investigate the association of the derived endotypes with chronic morbidity outcomes (recurrent wheeze by age 3 years and asthma at age 5 years) (Fig. 1). We report that four biologically distinct and clinically meaningful endotypes are identified: (A) clinical$^{classic}$microbiome$^{M.\ nonliquefaciens}$inflammation$^{IFN-intermediate}$, (B) clinical$^{atopic}$microbiome$^{S.\ pneumoniae/M.\ catarrhalis}$inflammation$^{IFN-high}$, (C) clinical$^{severe}$microbiome$^{mixed}$inflammation$^{IFN-low}$, and (D) clinical$^{non-atopic}$microbiome$^{M.catarrhalis}$inflammation$^{IL-6}$. Specifically, the endotype B characterized by a high proportion of parental asthma, immunoglobulin E (IgE) sensitization, and rhinovirus coinfection, S. pneumoniae/M. catarrhalis codominance, and high IFN-α and -γ response had the highest risk for developing asthma by age 5 years.

## Results
We analyzed the data from a multicentre prospective cohort study of infants hospitalized for bronchiolitis—the 35th Multicentre Airway Research Collaboration (MARC-35) study. This prospective cohort study completed enrollment of 1016 infants (age < 1 year) hospitalized with bronchiolitis at 17 sites across 14 U.S. states. Of these 1016 infants (median age, 3 months; female, 40%), 921 (91%) completed the run-in procedure (contact at both 1 week after hospital discharge or 3 weeks after hospitalization) and comprise the MARC-35 longitudinal cohort. Of the infants enrolled into this longitudinal cohort, this study included 221 infants with RSV bronchiolitis who were randomly selected for the nasopharyngeal microbiome, transcriptome, and metabolome testing (Supplementary Fig. 1). The analytic cohort and

nonanalytic cohort did not differ in patient characteristics ($P \geq$ 0.05; Supplementary Table 1), except for daycare use. Among the analytic cohort, the median age was 3 (IQR, 2–6) months, 42% were female, and 42% were non-Hispanic white. Overall, 72% were solo-RSV infection, while 13% had coinfection with rhinovirus. The analytic cohort had no patients lost to follow-up up to age 5 years.

**Integrated omics approach identified clinically- and biologically distinct endotypes.** To derive clinically- and biologically distinct RSV endotypes, we applied integrative network and clustering approaches to clinical, virus, nasopharyngeal airway microbiome, transcriptome, and metabolome data (Fig. 1). First, we computed a distance matrix of each dataset—(1) Gower distance for clinical and virus data (age, sex, birth weight, history of breathing problems, lifetime antibiotic use, parental asthma, IgE sensitization, positive pressure ventilation use, and virus data), (2) Bray–Curtis distance for microbiome data of 40 most abundant species, which accounted for 95% of total abundance, (3) Pearson distance for transcriptome data using 3000 gene transcripts with high variances, (4) Euclidian distance for metabolome data with using 100 metabolites with high variances with adjustment for batch effects. Then, we computed an affinity matrix of each dataset separately and derived a fused affinity matrix by a similarity network fusion. Lastly, to identify mutually exclusive endotypes, we applied spectral clustering to the fused affinity matrix. To choose the optimal number of endotypes, we used a combination of the average silhouette scores, network modularity, endotype size, and clinical and biological plausibility. Across the different numbers of endotypes ($k$ of 2–6), both the average silhouette score and the network modularity were highest with $k =$ 4 (Supplementary Fig. 2).

We identified four distinct endotypes among infants with RSV bronchiolitis (Supplementary Fig. 3). These endotypes were chiefly characterized by their clinical presentation, major bacteria species of the nasopharyngeal airway microbiome, and immune response: (A) clinical$^{classic}$microbiome$^{M.\ nonliquefaciens}$inflammation$^{IFN-intermediate}$ (19.5%), B) clinical$^{atopic}$microbiome$^{S.\ pneumoniae/M.\ catarrhalis}$ inflammation$^{IFN-high}$ (28.5%), C) clinical$^{severe}$microbiome$^{mixed}$inflammation$^{IFN-low}$ (28.5%), and D) clinical$^{non-atopic}$microbiome$^{M.\ catarrhalis}$inflammation$^{IL-6}$ (23.5%) (Figs. 2–5, Supplementary Figs. 4, 5 and Table 1). Across these endotypes, several clinical characteristics (e.g., age, lifetime antibiotic use, parental history of asthma, use of positive pressure ventilation during hospitalization) were significantly different ($P < 0.05$; Table 1).

Descriptively, infants with an endotype A were characterized by "classic" clinical presentation of bronchiolitis (e.g., young age, a low proportion of previous breathing problems and parental asthma, the high proportion of solo-RSV infection), higher abundance of M. nonliquefaciens, and intermediate IFN-α and -γ response (Table 1 and Figs. 2–5). Endotype B was characterized by a high proportion of parental asthma, IgE sensitization, and coinfection with rhinovirus, higher abundance of S. pneumoniae and M. catarrhalis, and higher IFN-α and -γ response (Table 1 and Figs. 2–5). Endotype C was characterized by a high proportion of lifetime antibiotics use and positive pressure ventilation use, mixed microbiome profile, and low IFN response (Figs. 2–4 and Supplementary Fig. 4). Lastly, the endotype D was characterized by a low proportion of parental asthma and IgE sensitization, high abundance of M. catarrhalis, and high interleukin-6 (IL-6) response profile (Figs. 2–4 and Supplementary Fig. 5). These variables that characterized the endotypes also had high-ranked normalized mutual information scores, indicating large contributions to the similarity network (Supplementary Fig. 6).

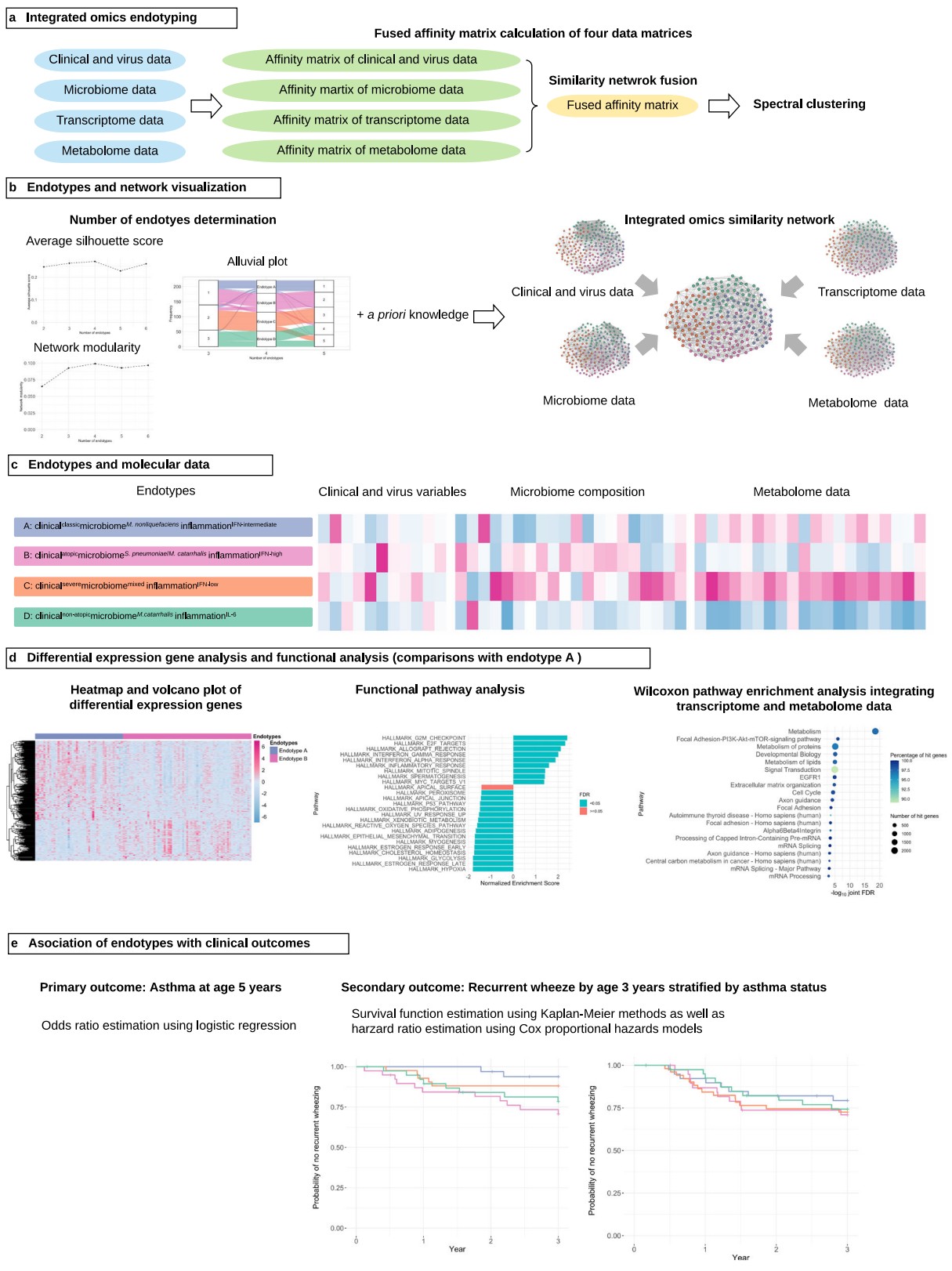

**Endotypes had differential risks of developing asthma and recurrent wheeze**. To examine their longitudinal association with clinical outcomes, we compared the outcome risks between endotype A (clinically "classic" bronchiolitis) and each of the other endotypes. Compared with endotype A infants, endotype B infants (clinical^atopic microbiome^S. pneumoniae/M. catarrhalis inflammation^IFN-high) had a significantly higher risk of developing asthma by age 5 years (9.3% vs. 38.1% [binary outcome]; OR, 6.00; 95% CI, 2.08–21.9; $P = 0.002$; Table 2) while those with an endotype C or D did not have significantly differential risks.

**Fig. 1 Analytic workflow of integrated omics endotyping. a** After an affinity matrix of each dataset (clinical and virus, microbiome, transcriptome, and metabolome) was separately computed, and a fused affinity matrix was generated by similarity network fusion. Then, the fused affinity matrix was used to identify mutually exclusive endotypes by spectral clustering. **b** A combination of average silhouette scores, network modularity, and clinical plausibility (in addition to endotype size) was used to choose the optimal number of endotypes. The concordance between the different numbers of endotypes was also examined. After deriving endotypes, a similarity network was visualized. **c** Between four derived endotypes of RSV bronchiolitis, the differences in the major clinical and virus variables, nasopharyngeal microbiome, and metabolome were visualized using heatmap. **d** Differentially expressed genes (endotype A as the reference group) were visualized using a heatmap and volcano plot. The functional pathway analysis using the gene set enrichment analysis and the Wilcoxon pathway enrichment analysis integrating transcriptomic and metabolome data were conducted to identify enriched pathways. **e** The risk of childhood asthma (binary outcome) was modeled by fitting a logistic regression model. The rate of recurrent wheeze (time-to-event outcome) was modeled by fitting a Cox proportional hazards model. RSV respiratory syncytial virus, IFN interferon, IL interleukin.

Likewise, in the rate of developing recurrent wheeze by age 3 years that resulted in asthma, the Kaplan–Meier curves significantly differed between the endotypes ($P_{\text{log-rank}} = 0.049$; Fig. 6). Compared with endotype A infants, endotype B infants had a significantly higher rate of recurrent wheeze that resulted in asthma (6.1% vs. 28.2%; HR, 5.50; 95% CI, 1.22–24.8; $P = 0.03$; Table 2). By contrast, for the rate of recurrent wheeze that did not result in asthma, there were no significant differences between endotypes A and B (20.5% vs. 28.2%; HR, 1.47; 95% CI, 0.59–3.66; $P = 0.40$; Table 2).

**Endotypes had distinct biological characteristics**. To better understand the difference between the lowest-risk (endotype A) and highest-risk (endotype B) groups, we compared the nasopharyngeal microbiome, transcriptome, and metabolome in both individual and integrated manner. For example, compared with endotype A infants, endotype B infants had a higher abundance of *S. pneumoniae* but the lower abundance of *M. nonliquefaciens* (both $P < 0.05$, FDR < 0.1; Supplementary Table 3). Similarly, the overall host transcriptome profile was different between these endotypes (Fig. 5a heatmap) with 29 differentially expressed genes (FDR < 0.1 with ≥|1.5|-fold change; Fig. 5a volcano plot; Supplementary Table 4). In the functional pathway analysis that identifies biologically meaningful pathways, endotype B infants had 24 differentially enriched pathways (FDR < 0.05)—e.g., upregulated IFN-α and -γ pathways (Fig. 5b). The integrated analysis of both transcriptome and metabolome data also demonstrated 155 differentially enriched pathways (FDR < 0.05) —e.g., enriched PI3K-Akt-mTOR-signaling pathway in endotype B (Fig. 4c). For the endotype A vs. C and endotype A vs. D comparisons, the detailed differences in transcriptome and integrated pathway analysis results are also summarized in Supplementary Figs. 4 and 5.

**Sensitivity analyses demonstrate the robustness of endotype-outcome associations**. In the sensitivity analyses that examine the robustness of primary findings, we first excluded infants with rhinovirus coinfection. Despite reduced statistical power, compared to endotype A, endotype B infants had a significantly higher risk of asthma (10.5% vs. 38.0%; OR, 5.21; 95% CI, 1.73–19.5; $P = 0.006$; Supplementary Table 5) and a nonsignificant but consistently higher rate of recurrent wheeze that resulted in asthma (7.14% vs. 28.1%; HR, 4.61; 95% CI, 0.99–21.3; $P = 0.051$; Supplementary Table 5). Second, we repeated the analyses without excluding the preprocessed variables. The alluvial plot (Supplementary Fig. 7) demonstrated consistency between the four endotypes derived in this sensitivity analysis and the original four endotypes (A–D). Compared to infants with an endotype 1 (which corresponds to endotype A), those with an endotype 2 (which corresponds to endotype B) had a significantly higher risk of asthma (14.6% vs. 36.4%; OR, 3.35; 95% CI, 1.35–9.19; $P = 0.012$; Supplementary Table 6). Lastly, we

examined different numbers of endotypes. The alluvial plot (Supplementary Fig. 8) demonstrated the consistency of the original four endotypes (A–D) across the different numbers. With the use of five endotypes, endotype 1 had 90% concordance with the original endotype A and endotype 2 had 100% concordance with the original endotype B (Supplementary Table 7). Similar to the primary analysis, these five endotypes were also characterized by clinical presentation and nasopharyngeal microbiome composition (e.g., *S. pneumoniae, M. catarrhalis*). In addition, the differential gene expression analysis demonstrated that, compared to endotype 1 (which is concordant with endotype A), endotype 2 (which is concordant with endotype B) infants also had upregulated IFN-γ and NFκB pathways (both FDR < 0.05; Supplementary Fig. 9). Lastly, compared to endotype 1, endotype 2 infants had a significantly higher risk of asthma at age 5 years (12.5% vs. 40.8%; OR, 4.83; 95% CI, 1.71–16.0; $P = 0.005$; Supplementary Table 8) and nonsignificant but consistently higher rate of recurrent wheeze that resulted in asthma (9.7% vs. 31.0%; HR, 3.57; 95% CI, 0.97–13.2; $P = 0.057$; Supplementary Table 8).

**Discussion**

By integrating clinical, virus, nasopharyngeal microbiome, transcriptome, and metabolome data from a multicentre prospective cohort study of 221 infants with RSV bronchiolitis, we identified four biologically distinct endotypes. In particular, compared to infants with endotype A ("classic" bronchiolitis), those with endotype B—characterized by a high proportion of parental asthma, IgE sensitization, and rhinovirus coinfection, *S. pneumoniae/M. catarrhalis* codominance, and high IFN-α and -γ response—had a significantly higher risk for developing childhood asthma. To the best of our knowledge, this is the first study that has identified biologically meaningful endotypes in infants with RSV bronchiolitis and demonstrated their longitudinal relations with the risk of chronic morbidities.

Recent studies have examined the potential mechanisms that link infant bronchiolitis to its health sequelae. For example, we previously reported that interactions between infecting virus and IgE sensitization in infants with bronchiolitis are associated with the risk of developing asthma[2,5]. Similarly, studies have also reported the association of unique airway microbiome profiles with the risk of recurrent wheeze and childhood asthma. Indeed, research has shown the pathogenic role of *Streptococcus* genus and *S. pneumoniae* in the upper airway—both among infants with or without bronchiolitis[13,14]—in the development of these chronic morbidities. However, the current literature has conflicting findings about the role of *Moraxella* genus in the infant airway, with higher *Moraxella* abundance associated with increased and decreased risk of developing asthma. In contrast to these earlier reports based on a culture-dependent method[13] or 16S rRNA gene sequencing[14,15], our study with the species-level resolution of microbiome structure revealed that the endotypes with distinct *Moraxella* species (less-pathogenic *M. nonliquefaciens*[16] vs. more-pathogenic *M. catarrhalis*[13]) exhibit

**Table 1 Baseline characteristics and clinical course of infants, according to respiratory syncytial virus bronchiolitis endotypes.**

| Characteristics | Overall (n = 221; 100%) | Endotype A (n = 43; 19.5%) | Endotype B (n = 63; 28.5%) | Endotype C (n = 63; 28.5%) | Endotype D (n = 52; 23.5%) | P value* |
|---|---|---|---|---|---|---|
| *Demographics* | | | | | | |
| Age (month), median (IQR) | 3 (2–6) | 2 (1–4) | 4 (2–6) | 4 (2–8) | 3 (1–4) | 0.002 |
| Female sex | 92 (41.6) | 35 (81.4) | 30 (47.6) | 20 (31.7) | 7 (13.5) | <0.001 |
| Race/ethnicity | | | | | | 0.40 |
| Non-Hispanic white | 92 (41.6) | 21 (48.8) | 17 (27.0) | 32 (50.8) | 22 (42.3) | |
| Non-Hispanic black | 54 (24.4) | 9 (20.9) | 20 (31.7) | 12 (19.0) | 13 (25.0) | |
| Hispanic | 66 (29.9) | 12 (27.9) | 22 (34.9) | 17 (27.0) | 15 (28.8) | |
| Other or unknown | 9 (4.1) | 1 (2.3) | 4 (6.3) | 2 (3.2) | 2 (3.8) | |
| Prematurity (32–37 weeks) | 46 (20.8) | 7 (16.3) | 14 (22.2) | 10 (15.9) | 15 (28.8) | 0.33 |
| Birth weight (kg), median (IQR) | 3.20 (2.85–3.54) | 3.14 (2.92–3.42) | 3.02 (2.81–3.40) | 3.30 (2.92–3.58) | 3.30 (2.68–3.64) | 0.36 |
| Mode of birth (cesarean delivery) | 75 (34.6) | 13 (30.2) | 19 (30.6) | 23 (36.5) | 20 (40.8) | 0.63 |
| Previous breathing problems (count) | | | | | | 0.26 |
| 0 | 189 (85.5) | 38 (88.4) | 58 (92.1) | 48 (76.2) | 45 (86.5) | |
| 1 | 24 (10.9) | 3 (7.0) | 4 (6.3) | 11 (17.5) | 6 (11.5) | |
| 2 | 8 (3.6) | 2 (4.7) | 1 (1.6) | 4 (6.3) | 1 (1.9) | |
| Previous ICU admission | 4 (1.8) | 0 (0) | 0 (0) | 3 (4.8) | 1 (1.9) | 0.21 |
| Lifetime antibiotic use[†] | 67 (30.3) | 2 (4.7) | 12 (19.0) | 48 (76.2) | 5 (9.6) | <0.001 |
| Ever attended daycare | 66 (29.9) | 13 (30.2) | 21 (33.3) | 18 (28.6) | 14 (26.9) | 0.89 |
| Cigarette smoke exposure at home | 32 (14.5) | 11 (25.6) | 11 (17.5) | 6 (9.5) | 4 (7.7) | 0.06 |
| Maternal smoking during pregnancy | 30 (13.8) | 8 (18.6) | 6 (9.7) | 10 (15.9) | 6 (12.2) | 0.56 |
| Parental history of asthma | 68 (30.8) | 6 (14.0) | 50 (79.4) | 10 (15.9) | 2 (3.8) | <0.001 |
| Parental history of eczema | 41 (18.6) | 9 (20.9) | 14 (22.2) | 13 (20.6) | 5 (9.6) | 0.27 |
| *Clinical presentation* | | | | | | |
| Weight (kg), median (IQR) | 5.90 (4.60–7.90) | 4.90 (4.14–5.65) | 6.20 (5.18–7.75) | 6.73 (4.90–8.35) | 5.64 (4.40–7.26) | 0.003 |
| Respiratory rate (per minute), median (IQR) | 48 (40–60) | 52 (41–61) | 48 (40–62) | 45 (38–53) | 52 (44–60) | 0.045 |
| Oxygen saturation | | | | | | 0.28 |
| <90% | 27 (12.6) | 5 (11.9) | 4 (6.3) | 12 (20.3) | 6 (11.8) | |
| 90–93% | 173 (80.5) | 35 (83.3) | 55 (87.3) | 44 (74.6) | 39 (76.5) | |
| ≥94% | 15 (7.0) | 2 (4.8) | 4 (6.3) | 3 (5.1) | 6 (11.8) | |
| Blood eosinophilia (≥4%) | 18 (9.7) | 5 (13.5) | 5 (9.8) | 3 (5.8) | 5 (11.1) | 0.64 |
| IgE sensitization | 46 (20.8) | 7 (16.3) | 15 (23.8) | 16 (25.4) | 8 (15.4) | 0.48 |
| *Clinical course* | | | | | | |
| Positive pressure ventilation use[‡] | 17 (7.7) | 2 (4.7) | 3 (4.8) | 12 (19.0) | 0 (0) | 0.001 |
| Intensive treatment use[§] | 37 (16.7) | 6 (14.0) | 10 (15.9) | 16 (25.4) | 5 (9.6) | 0.15 |
| Length-of-day (day), median (IQR) | 2 (1–3) | 2 (1–3) | 2 (1–4) | 2 (1–4) | 2 (1–3) | 0.17 |
| Antibiotic use during hospitalization | 71 (32.1) | 9 (20.9) | 21 (33.3) | 32 (50.8) | 9 (17.3) | <0.001 |
| Corticosteroid use during hospitalization | 24 (10.9) | 1 (2.3) | 6 (9.5) | 13 (20.6) | 4 (7.7) | 0.02 |
| *Respiratory virus* | | | | | | |
| RSV solo infection | 158 (71.5) | 32 (74.4) | 44 (69.8) | 40 (63.5) | 42 (80.8) | 0.22 |
| Rhinovirus coinfection | 29 (13.1) | 5 (11.6) | 13 (20.6) | 5 (7.9) | 6 (11.5) | 0.21 |
| Rhinovirus-A | 14 (6.3) | 1 (2.3) | 7 (11.1) | 5 (7.9) | 1 (1.9) | 0.15 |
| Rhinovirus-B | 4 (1.8) | 2 (4.7) | 1 (1.6) | 0 (0.0) | 1 (1.9) | 0.37 |
| Rhinovirus-C | 11 (5.0) | 2 (4.7) | 5 (7.9) | 0 (0.0) | 4 (7.7) | 0.09 |
| Other coinfection pathogens[‖] | 34 (15.4) | 6 (14.0) | 6 (9.5) | 18 (28.6) | 4 (7.7) | 0.006 |
| *Chronic comorbidities* | | | | | | |
| Asthma at age 5 years | 51 (23.1) | 4 (9.3) | 24 (38.1) | 12 (19.0) | 11 (21.2) | 0.005 |
| Recurrent wheeze by age 3 years | 69 (31.2) | 10 (23.3) | 22 (34.9) | 19 (30.2) | 18 (34.6) | 0.58 |

*IQR* interquartile range, *ICU* intensive care unit, *RSV* respiratory syncytial virus, *IgE* immunoglobulin E.
Data are no. (%) of infants unless otherwise indicated. Percentages may not equal 100, because of rounding and missingness.
*Two-sided raw *P* values.
[†]Any systemic antibiotic use from birth up to the index hospitalization for bronchiolitis.
[‡]Infants with bronchiolitis who underwent continuous positive airway ventilation and/or mechanical ventilation.
[§]Infants with bronchiolitis who were admitted to ICU and/or who underwent positive pressure ventilation.
[‖]Infants with coinfection by non-rhinovirus include adenovirus infection (n = 7), bocavirus (n = 8), endemic coronavirus (n = 15), enterovirus (n = 1), influenza virus (n = 1), human metapneumovirus (n = 4), *Mycoplasma pneumonia* (n = 1) and parainfluenza virus (n = 3). Since six infants have coinfection with multiple infecting agents, the total number is not equal to 34.

differential risks of chronic morbidities. Research that examined the role of airway immune response in RSV bronchiolitis and its sequalae has also shown somewhat conflicting results—e.g., the associations of higher type-2/-17 cytokines[10] or CCL-5 chemokine[11] with higher risks of recurrent wheeze. Moreover, research has also suggested the role of metabolome—which represents the downstream functional products of the microbiome, child's genetic make-up, and immune response—in the pathogenesis of bronchiolitis and asthma[2]. For example, a recent study of airway metabolome has shown differences in pro- and

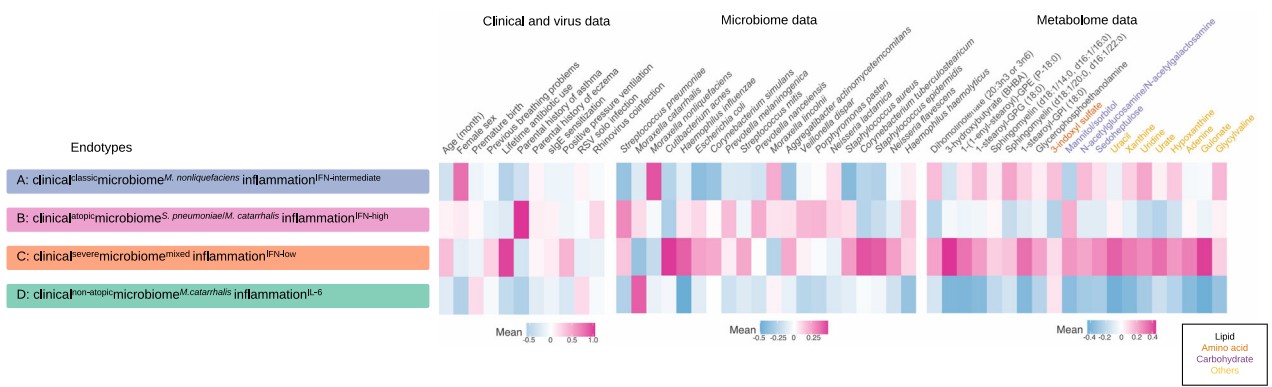

**Fig. 2 Between-endotype differences in clinical variables, virus, nasopharyngeal microbiome, and nasopharyngeal metabolome data in infants with respiratory syncytial virus bronchiolitis.** To visualize the between-endotype differences, the clinical variables and viruses are treated as numeric variables and processed by autoscaling. The microbiome data (20 most abundant species) are processed by $\log_2$ transformation and autoscaling. The metabolome data (20 metabolites with the highest normalized mutual information score) are processed by $\log_2$ transformation with batch effect adjustment and autoscaling. RSV respiratory syncytial virus, IFN interferon, IL interleukin, sIgE specific immunoglobulin E.

anti-inflammatory mediators (e.g., spermidine) in infants with bronchiolitis[12]. These studies—while limited to single-level data (e.g., clinical, virus, microbiome, cytokines/chemokines, metabolome)—have discovered potential mechanisms that underlie the bronchiolitis–asthma link. The current study corroborates these earlier reports and extends them by identifying biologically meaningful RSV bronchiolitis endotypes that have differential risks of chronic morbidities.

There are several potential mechanisms linking the endotypes of RSV bronchiolitis—in particular endotype B (clinical[atopic]microbiome[S. pneumoniae/M. catarrhalis]inflammation[IFN-high])—with subsequent chronic morbidities. Emerging evidence suggests that RSV infection increases the virulence of S. pneumoniae[17] and induces type I IFN production by macrophages in S. pneumoniae infection[18]. Likewise, airway infection of mice by M. catarrhalis induces an inflammatory response with CD4+ T-cell-derived IFN-γ[19]. In addition, a recent study has shown that the higher types 1 and 3 IFN response of RSV-infected bronchial epithelial cells from children with asthma is correlated with the magnitude of airway obstruction, suggesting that an accelerated airway IFN response to RSV infection promotes the obstructive processes of lungs[20]. Furthermore, in an animal model of allergic asthma, there were synergistic effects of pneumovirus infection and allergic sensitization on type 1 IFN response[21]. In this study, the integrated analysis also demonstrated that endotype B had upregulated PI3K-Akt-mTOR-signaling pathway. The literature has shown its role in the pathobiology of asthma, including activating both innate and adaptive immunity, and airway remodeling[22]. PI3K inhibition not only reduces allergen-induced inflammation and hyperresponsiveness but also prevents the expression of IFN-γ-induced protein 10—a mediator released by virus-induced asthma[22]. Besides, both the PI3K-Akt-mTOR-signaling pathway and anti-inflammatory polyamines regulate the function of dendritic cells linking innate to adaptive immunity[23]. In conjunction with earlier studies, our data collectively suggest the integrated role of RSV infection, pathogenic bacteria (S. pneumoniae and M. catarrhalis), and IFN-α/-γ airway response in the development of asthma among infants at risk for atopy.

In addition to endotype B, the identification of other endotypes such as endotype C—characterized by a high proportion of previous antibiotic use, higher bronchiolitis severity, mixed microbiome profile, and low IFN-α/-γ response—is also intriguing. Consistently, studies have reported the relations of early-life exposures to systematic antibiotics, airway dysbiosis, and higher asthma risk[15] and

the association of immature IFN-α and -γ response in RSV infection with severe RSV infection[24] contributing to direct damage to airway structure and subsequent risk of asthma[25]. Lastly, IL-6, which partially characterized endotype D (clinical[non-atopic] microbiome[M. catarrhalis]inflammation[IL-6]), also plays a role in the pathobiology of both severe respiratory infection and asthma. In agreement with our study, research has previously shown the interrelations between RSV infection, M. catarrhalis, higher nasopharyngeal IL-6 level, and wheezing[19,26]. Notwithstanding the complexity of these mechanisms, the observed endotypes of RSV bronchiolitis and their longitudinal relations with chronic morbidities are important findings that could advance research into the development of endotype-specific strategies for bronchiolitis treatment and asthma prevention.

Our study has several potential limitations. First, we focused on infants with RSV bronchiolitis while other non-RSV viruses are also causative pathogens. However, RSV not only contributes to 75% of severe bronchiolitis[7] but also has the largest population attributable fraction in asthma development[3]. Second, bronchiolitis involves inflammation of the lower airways in addition to the upper airways. While our study is based on nasopharyngeal samples, studies have shown that upper airway sampling provides a reliable representation of the lung microbiome[27] and transcriptome[28] profiles. Furthermore, the use of upper airway specimens is preferable as bronchoscopy or other methods of lower airway sampling would be too invasive in young infants. Third, the nasopharyngeal samples were obtained at a single time point. While longitudinal molecular data are also important, the study objective was to identify endotypes of RSV bronchiolitis. Besides, even with single-time point data, we successfully identified biologically distinct endotypes that are longitudinally associated with chronic morbidities. Fourth, it is possible that asthma diagnosis (by age 5 years) is misclassified and that children are going to develop asthma at a later state. To address these points, the study sample is currently being followed longitudinally up to age 9 years. Fifth, this study did not have healthy "controls". Yet, the study objective was not to evaluate the difference of endotypes from healthy infants but to define endotypes of RSV bronchiolitis. Sixth, the sample size of the present analysis is relatively smaller to the number of variables examined. This study should facilitate further validation research. Lastly, the study sample consisted of racially/ethnically- and geographically diverse infants hospitalized for bronchiolitis. While our sample had a large severity contrast, our inferences may not be generalizable to infants in ambulatory settings with mild-to-moderate

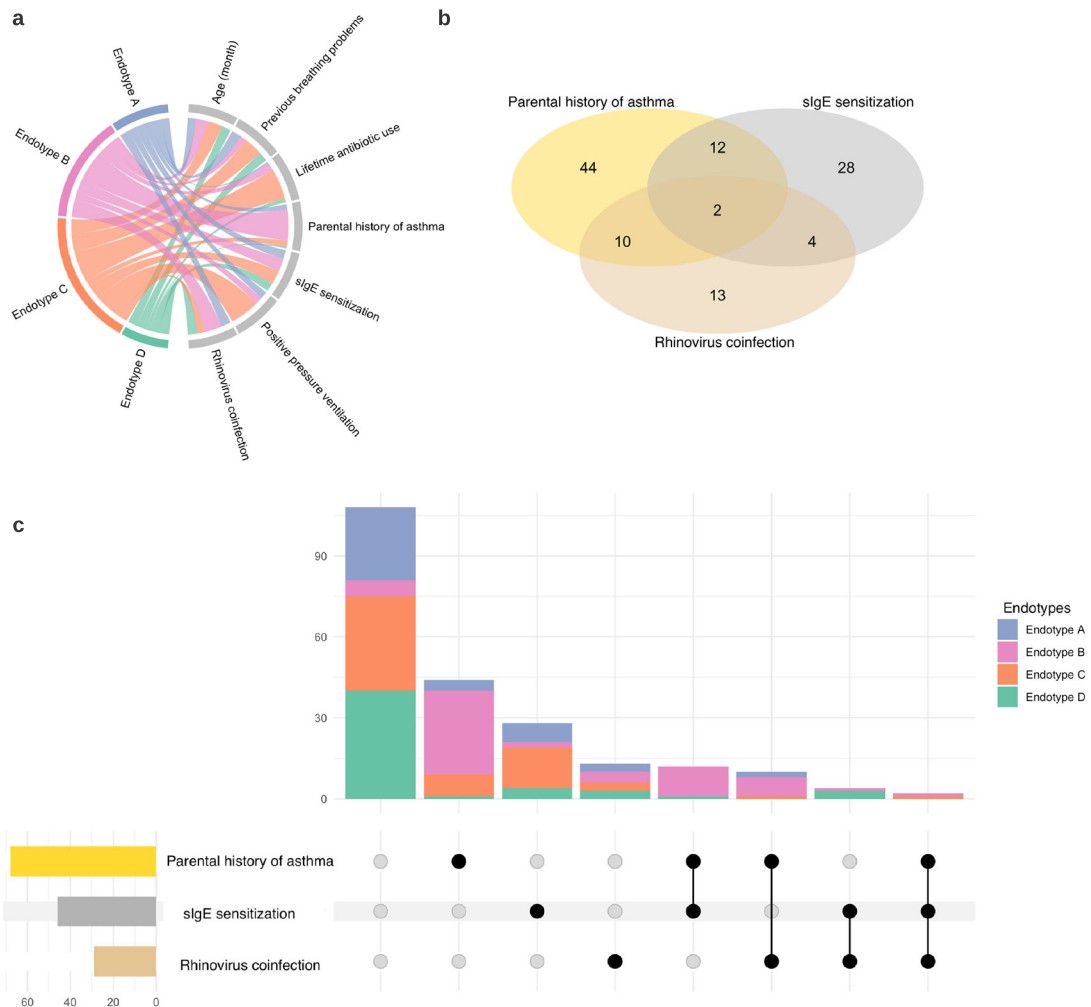

**Fig. 3 Relationship between major clinical variables and endotypes. a** Chord diagram showing major clinical variables by endotype. The ribbons connect from the individual endotypes to the major clinical and virus characteristics. The width of the ribbon represents the proportion of infants within the endotype who have the corresponding clinical or virus characteristic. Then, it was scaled to a total of 100%. For example, the endotype B infants (light red) had a high proportion of parental asthma, IgE sensitization, and coinfection with rhinovirus. Endotype C (light orange) infants had a high proportion of lifetime antibiotics use and positive pressure ventilation use during the index hospitalization for bronchiolitis. **b** Venn diagram of three major clinical variables (parental history of asthma, IgE sensitization, rhinovirus infection) and their intersections. The Venn diagram illustrates the composition of three major clinical variables and their intersections. The numbers correspond to the number of infants in each subset and intersection. **c** Upset plot corresponding to the presented Venn diagram. The plot illustrates the composition of three major clinical variables and their intersections visualized based on the four endotypes. Vertical stacked bar charts reflect the number of infants within each subset and intersection colored according to the endotypes. Horizontal bars indicate the number of infants in each clinical variable set. Black dots indicate the sets of subsets and intersections; connecting lines indicate relevant intersections related to each stacked bar chart.

bronchiolitis and warrant external and experimental validation. Nonetheless, our data remain relevant for the 110,000 infants hospitalized yearly in the U.S.[1], a vulnerable population with a substantial morbidity burden.

In summary, by applying an integrated omics approach to data from a multicentre prospective cohort study of 221 infants with RSV bronchiolitis, we identified four biologically distinct and clinically meaningful endotypes. Specifically, the endotype characterized by a high proportion of parental asthma, IgE sensitization, and rhinovirus coinfection, *S. pneumoniae*/*M. catarrhalis* codominance, and high IFN-α and -γ response had the highest risk for developing asthma in later childhood. Considering the relatively small sample size of infants with severe RSV bronchiolitis in the present study, external validation is warranted. Regardless, our data lend significant support to the emerging concept that "bronchiolitis" represents several diseases with distinct biological mechanisms. For clinicians, our findings may provide an evidence base for the early identification of high-risk

children during an important period of airway development—early infancy. For researchers, our data should facilitate further investigations into the development of endotype-specific strategies for bronchiolitis treatment and asthma prevention.

## Methods

**Ethical statements**. With the exception of specimen collection (the NPA, blood, and nasal swab), all study participants were evaluated and treated as usual and without regard to this observational study. Parent/legal guardians were approached about participating after the medical team had finished their assessments and stabilized the study participant. If necessary, the recruiting took place the morning after admission, but no later than 24 h after admission to the ward or intensive care unit. Samples and information of the participants were used to study the possible genetic causes of severe bronchiolitis, recurrent wheezing, asthma, and related concepts. The institutional review board at each of the participating hospitals approved the study. Written informed consent was obtained from the parent or guardian.

**Study design, setting, and participants**. We analyzed data from a multicentre prospective cohort study of infants hospitalized for bronchiolitis—the 35th Multicentre Airway Research Collaboration (MARC-35) study[7]. MARC-35 is

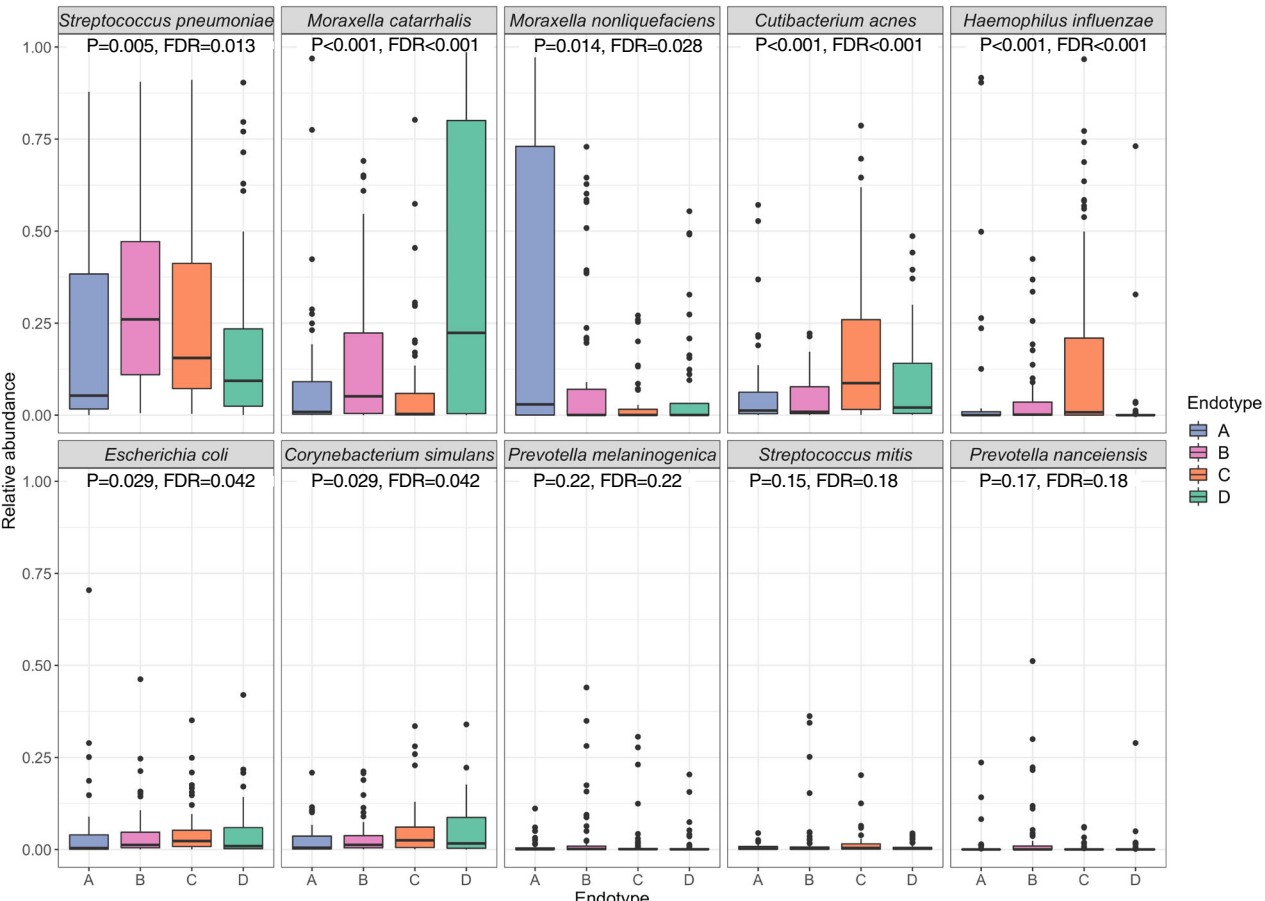

**Fig. 4 Between-endotype differences in the abundance of the ten most abundant nasopharyngeal microbial species.** The boxplots show the distribution of the ten most abundant microbial species of the nasopharyngeal microbiome, according to the four endotypes. The differences in relative abundance (scale of 0–1) among the four endotypes (endotype A, 43 samples; endotype B, 63 samples; endotype C, 63 samples; and endotype D, 52 samples) were tested by Kruskal–Wallis test. center lines indicate median values. Box limits indicate upper and lower quartiles. Whiskers indicate 1.5 × interquartile ranges. Points indicate outliers. Exact *P* values and false discovery rates (FDRs) are the following: *Streptococcus pneumoniae*, *P* value = 0.0051, FDR = 0.013; *Moraxella catarrhalis*, *P* value = $4.1 \times 10^{-5}$, FDR = 0.00014; *Moraxella nonliquefaciens*, *P* value = 0.014, FDR = 0.028; *Cutibacterium acnes*; *P* value = $2.4 \times 10^{-5}$, FDR = 0.00012; *Haemophilus influenzae*, *P* value = $2.0 \times 10^{-5}$, FDR = 0.00012; *Escherichia coli*, *P* value = 0.029, FDR = 0.042; *Corynebacterium simulans*, *P* value = 0.029, FDR = 0.042; *Prevotella melaninogenica*, *P* value = 0.22, FDR = 0.22; *Streptococcus mitis*, *P* value = 0.15, FDR = 0.18; and *Prevotella nanceiensis*, *P* value = 0.17, FDR = 0.18. FDR false discovery rate.

coordinated by the Emergency Medicine Network (EMNet,), an international research collaboration with 247 participating hospitals. Site investigators enrolled infants (age < 1 year) hospitalized with bronchiolitis at 17 sites across 14 U.S. states using a standardized protocol during three consecutive bronchiolitis seasons (from November 1 through April 30) during 2011–2014. The diagnosis of bronchiolitis was made according to the American Academy of Pediatrics bronchiolitis guidelines, defined as the acute respiratory illness with a combination of rhinitis, cough, tachypnoea, wheezing, crackles, or retraction[29]. We excluded infants with a preexisting heart and lung disease, immunodeficiency, immunosuppression, or gestational age of <32 weeks, history of previous bronchiolitis hospitalization, or those who were transferred to a participating hospital >24 h after initial hospitalization. Of 921 infants enrolled in the longitudinal cohort, the current analysis investigated 221 infants with RSV infection who were randomly selected for the nasopharyngeal microbiome, host transcriptome, and metabolome testing (Supplementary Fig. 1).

**Data collection and measurement of virus, microbiome, transcriptome, and metabolome.** Clinical data (patients' demographic characteristics, and family, environmental, and medical history, and details of the acute illness) were collected via structured interview and chart reviews[7]. All data were reviewed at the EMNet Coordinating Centre (Boston, MA), and site investigators were queried about missing data and discrepancies identified by manual data checks. In addition to the clinical data, nasopharyngeal airway samples were collected by trained site investigators using the standardized protocol that was utilized in a previous cohort study of children with bronchiolitis[7,30]. All sites used the same collection equipment (Medline Industries, Mundelein, IL, USA) and collected the samples within 24 h of

hospitalization. The nasopharyngeal airway specimens were immediately placed on ice and then stored at −80 °C. Frozen specimens were shipped in batches to Baylor College of Medicine (Houston, TX) where they were tested for 17 respiratory viruses (including respiratory syncytial virus [RSV]) using real-time polymerase chain reaction (RT-PCR) assays (Supplementary Table 9)[7,30,31]. Frozen samples were also shipped to the University of Maryland (Baltimore, MD) for RNA sequencing (both transcriptome and metatranscriptome profiling) and to Metabolon (Durham, NC, USA) for metabolomic profiling[32].

*RNA extraction, RNA sequencing, and quality control.* Total RNA was isolated from the nasopharyngeal samples using Trizol LS reagent (ThermoFisher Scientific, Waltham, MA) in combination with the Direct-zol RNA Miniprep Kit (Zymo Research, Irvine, CA). RNA quantity was measured with the Qubit 2.0 fluorometer (ThermoFisher Scientific, Waltham, MA); its quality was assessed with the Agilent Bioanalyzer 2100 (Agilent, Palo Alto, CA) using the RNA 6000 Nano kit. Total RNA underwent DNase treatment using the TURBO DNA-free™ Kit (Thermo-Fisher Scientific, Waltham, MA) and rRNA reduction for both human and bacterial rRNA using NEBNext rRNA Depletion Kits (New England Biolabs, Ipswich, MA). RNA was prepared for sequencing using the NEBNext Ultra II Directional RNA Library Prep Kit (New England Biolabs, Ipswich, MA) and sequenced on an Illumina NovaSeq6000 using a S4 100PE Flowcell (Illumina, San Diego, CA). All RNAseq samples had sufficient sequence depth (mean, 8,067,019 pair-end reads/sample) to obtain a high degree of sequence coverage.

*Nasopharyngeal airway microbiome profiling.* Raw sequence reads were filtered and trimmed for adapters and contaminants using the *k*-mers strategy in bbduck[33] and

**Table 2 Association of respiratory syncytial virus bronchiolitis endotypes in infants with development of asthma and recurrent wheeze outcomes.**

| Endotypes | Childhood asthma at age 5 years* | | Recurrent wheeze by age 3 years† with asthma‡ | | Recurrent wheeze by age 3 years† without asthma§ | |
|---|---|---|---|---|---|---|
| | Odds ratio (95% CI) | P value‖ | Hazard ratio (95% CI) | P value‖ | Hazard ratio (95% CI) | P value‖ |
| Endotype A (clinical:classic microbiome:M. nonliquefaciens inflammation:IFN-intermediate) | 1 [Reference] | – | 1 [Reference] | – | 1 [Reference] | – |
| Endotype B (clinical:atopic microbiome:S. pneumoniae/M. catarrhalis inflammation:IFN-high) | 6.00 (2.08–21.9) | 0.002 | 5.50 (1.22–24.8) | 0.03 | 1.47 (0.59–3.66) | 0.40 |
| Endotype C (clinical:severe microbiome:Mixed inflammation:IFN-low) | 2.29 (0.74–8.07) | 0.18 | 2.67 (0.40–10.7) | 0.39 | 1.40 (0.59–3.34) | 0.45 |
| Endotype D (clinical:non-atopic microbiome:M. catarrhalis inflammation:IL-6) | 2.62 (0.82–10.1) | 0.12 | 3.86 (0.82–18.2) | 0.09 | 1.22 (0.48–3.10) | 0.67 |

CI confidence interval.
*Asthma (binary outcome) was defined as physician-diagnosis of asthma by age 5 years, plus either asthma medication use (e.g., albuterol inhaler, inhaled corticosteroids, montelukast) or asthma-related symptoms in the preceding year. To examine the association between RSV bronchiolitis endotypes (endotype A as the reference) and the risk of developing childhood asthma, the logistic regression model was fit.
†Recurrent wheeze (time-to-event outcome) was defined as having at least two corticosteroid-requiring exacerbations in 6 months or at least four wheezing episodes in 1 year that last at least 1 day and affect sleep. To examine the association between RSV bronchiolitis endotypes (endotype A as the reference) and the rate of recurrent wheeze, the Cox proportional hazards model was fit.
‡The outcome is recurrent wheeze by age 3 years with the epidemiological definition of asthma (n = 26) vs. no recurrent wheeze (n = 68). The analysis excludes the other children (n = 127).
§The outcome is recurrent wheeze by age 3 years without epidemiological definition of asthma (n = 43) vs. no recurrent wheeze (n = 51). The analysis excludes the other children (n = 127).
‖Two-sided raw P values.

default settings. We used PathoScope[34–36] and the expanded Human Oral Microbiome Database (eHOMD) database[37] to infer bacterial composition. Samples with <1000 reads, singletons, and strains not present in at least 10% of the samples were eliminated. The metatranscriptomic analysis obtained 1,968,352,599 merged sequences and identified 323 microbial lineages after singleton removal. The microbiome data are presented in Figs. 2, 4, and 8, and Supplementary Table 3.

*Nasopharyngeal airway host transcriptome profiling.* Transcript abundances from clean RNAseq reads were estimated in Salmon[38] using the human transcriptome (hg38) and the mapping-based mode. We first generated a decoy-aware transcriptome and then quantified the reads using Salmon's default settings and the following flags: –validateMappings, –recoverOrphans, –seqBias and –gcBias. Salmon is fast and accurate, corrects for potential changes in gene length across samples (e.g., from differential isoform usage), and has great sensitivity. For the differential gene expression analysis, Salmon's estimated transcript abundances were imported into DESeq2 using tximport[39]. The transcriptome data are presented in Fig. 5, Supplementary Table 3, and Supplementary Figs. 4, 5, and 9.

*Nasopharyngeal airway metabolomic profiling.* Metabolomic profiling used 125 µl of nasopharyngeal airway sample. All samples were blinded to Metabolon and processed in random order. The metabolic profiling used ACQUITY ultra-high performance liquid chromatography (UPLC) (Waters, Milford, MA, USA) and Q-Exactive high resolution/accurate MS interfaced with a heated electrospray ionization (HESI-II) source and Orbitrap mass analyzer operated at 35,000 mass resolution (ThermoFisher Scientific, Waltham, MA, USA).

Sample preparation was carried out as described previously[40,41]. In brief, recovery standards were added prior to the first step in the extraction process for quality control purposes. Proteins were precipitated with 500 µL of methanol added to 100 µL of the sample under vigorous shaking for 2 min (Glen Mills Genogrinder 2000; Clifton, NJ, USA) followed by centrifugation. The sample extract was dried then reconstituted in solvents compatible with each of the four methods. Each reconstitution solvent contained a series of standards at fixed concentrations to ensure injection and chromatographic consistency. One aliquot was analyzed using acidic positive ion conditions, chromatographically optimized for hydrophilic compounds. In this method, the extract was gradient eluted from a C18 column (Waters UPLC BEH C18-2.1 × 100 mm, 1.7 µm) using water and methanol, containing 0.05% perfluoropentanoic acid and 0.1% formic acid. Another aliquot was also analyzed using acidic positive ion conditions, but it was chromatographically optimized for hydrophobic compounds. In this method, the extract was gradient eluted from the same aforementioned C18 column using methanol, acetonitrile, water, 0.05% perfluoropentanoic acid, and 0.01% formic acid and was operated at an overall higher organic content. Another aliquot was analyzed using basic negative ion optimized conditions using a separate dedicated C18 column. The basic extracts were gradient eluted from the column using methanol and water, including 6.5 mM ammonium bicarbonate at pH 8. The fourth aliquot was analyzed via negative ionization following elution from a HILIC column (Waters UPLC BEH Amide 2.1 × 150 mm, 1.7 µm) using a gradient consisting of water and acetonitrile with 10 mM ammonium formate, pH 10.8. The MS analysis alternated between MS and data-dependent $MS^n$ scans using dynamic exclusion. The scan range varied slightly between methods but covered 70–1000 m/z. Parameters for chromatography are summarized in Supplementary Table 10.

Metabolites were identified by automated comparison of the ion features in the experimental samples to a reference library of chemical standard entries that include retention time, molecular weight (m/z), preferred adducts, and in-source fragments as well as associated MS spectra, and curated by visual inspection for quality control using QUICS software[42]. Identification of known chemical entities was based on comparisons to metabolomic library entries of >3000 purified standards. Peaks were quantified using the area under the curve. The raw area counts for each metabolite in each sample were normalized to correct for variation due to instrument interday tuning differences by the median value for each run day, setting the median to 1.0 for each run. Missing values were imputed with the observed minimum for that particular compound.

Four types of quality controls were analyzed in concert with the specimens: (1) samples generated from a pool from a small portion of each experimental specimen that served as technical replicate; (2) extracted water samples that served as process blanks; (3) samples of solvent used in extraction; and (4) a cocktail of standards spiked into every analyzed specimen that allowed instrument performance monitoring. The median relative standard deviation (RSD) for the standards that are added to each sample—a measure of instrument variability—was <5%. The metabolome data are presented in Figs. 2, 5, Supplementary Tables 2, 3, and 7, and Supplementary Figs. 4, 5, and 9.

**Outcome measures.** The primary outcome was asthma at age 5 years. The secondary outcome was the development of recurrent wheeze by age 3 years, stratified by asthma status to account for the heterogeneity of recurrent wheeze according to prior research[5]. The definition of asthma was based on a commonly used epidemiologic definition of asthma[43]—physician-diagnosis of asthma by age 5 years, plus either asthma medication use (e.g., albuterol inhaler, inhaled corticosteroids, montelukast) or asthma-related symptoms in the preceding year. The definition of recurrent wheeze was, based on the 2007

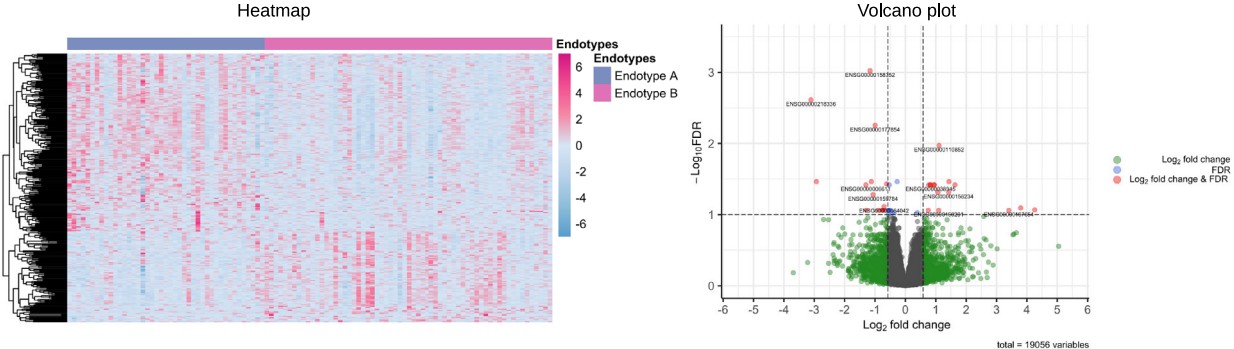

**Comparison of endotype A (reference) with endotype B**

**a** Heatmap and volcano plot of differentially-expressed genes

**b** Functional pathway analysis

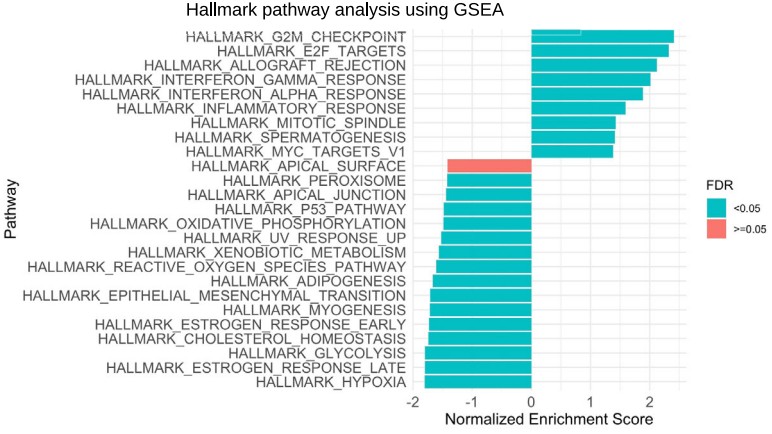

**c** Wilcoxon pathway enrichment analysis integrating transcriptome and metabolome data

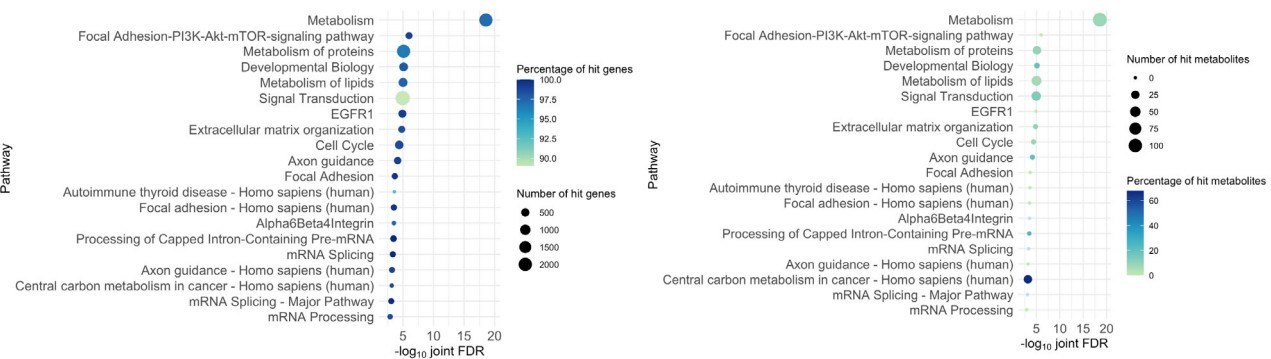

**Fig. 5 Differential gene expression analysis and functional pathway analysis in the endotypes A vs. B comparison. a** Heatmap and volcano plot of differentially expressed genes. For the heatmap (left), we selected 300 genes with the most significant *P* value (two-sided raw *P* value) and the color bar indicates the scaled value of variance stabilizing transformation. For the volcano plot (right), the threshold of log₂ fold change is |0.58| (i.e., ≥|1.5|-fold change) and that of FDR < 0.1. There were 29 differentially expressed genes that met these criteria. Of these, 11 Ensemble gene ids are annotated in the volcano plot due to space availability. These genes are presented in Supplementary Table 4. **b** Functional pathway analysis. For the functional class scoring analysis (left), we selected 25 pathways with the highest absolute value of normalized enriched score to visualize the plot. **c** Wilcoxon pathway enrichment analysis integrating transcriptome and metabolome data. For the Wilcoxon pathway enrichment analysis, we selected 20 pathways with the most significant joint FDR, and showed the numbers and proportions of hit genes (left) and metabolites (right) for the corresponding pathways. GSEA gene set enrichment analysis, FDR false discovery rate.

U.S. asthma guidelines[44], defined as having at least two corticosteroid-requiring exacerbations in 6 months or at least four wheezing episodes in 1 year that last at least 1 day and affect sleep.

**Statistical analysis**. In the current study, our aims are to identify biologically distinct RSV bronchiolitis endotypes (description [clustering]) and to relate them to outcome risks (association)[45]. The analytic workflow is summarized in Fig. 1. First, we selected and preprocessed variables to compute a distance matrix for each

of the datasets (i.e., clinical and virus, microbiome, transcriptome, and metabolome). For the clinical data, we choose variables based on a priori knowledge[3]—age, sex, birth weight, history of breathing problems, lifetime antibiotic use, parental asthma, immunoglobulin E (IgE) sensitization, positive pressure ventilation use, and virus data (coinfection with rhinovirus [binary], rhinovirus species (categorical), virus genomic load [count])[3,5]. The breathing problem variable had missingness in 2 infants (0.9%) which were imputed by a random forest method using R missForest package[46]. For the microbiome data, we used the relative abundance of 40 most abundant species which accounted for 95% of total abundance. For the

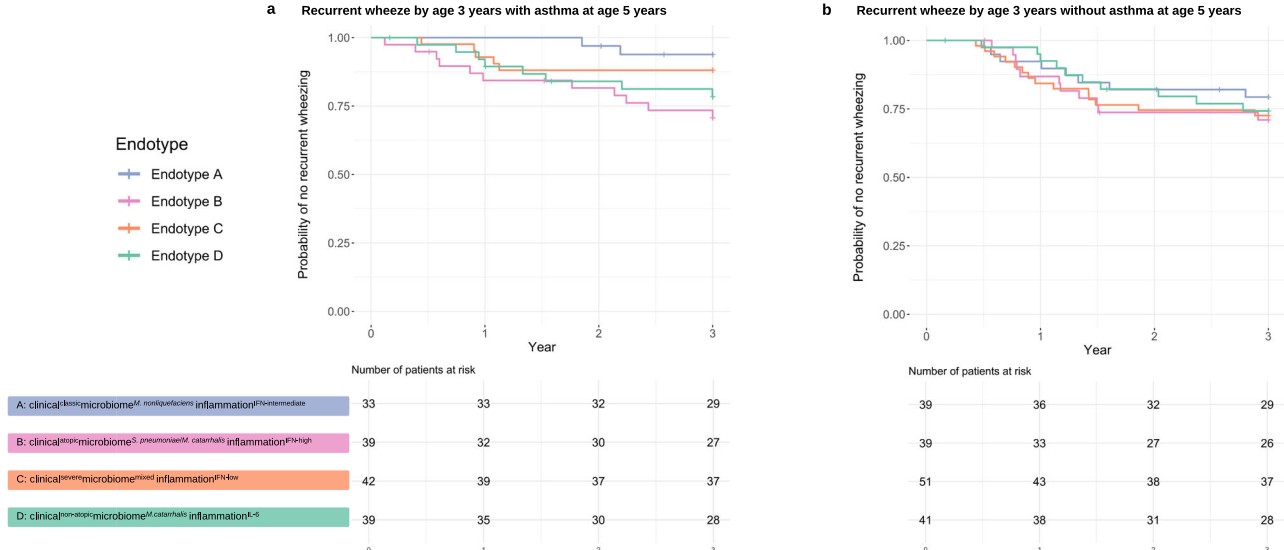

**Fig. 6 Kaplan–Meier curves for development of recurrent wheeze by age 3 years, according to respiratory syncytial virus bronchiolitis endotypes. a** Recurrent wheeze by age 3 years with asthma at age 5 years. Overall, the survival curves significantly differed across the endotypes ($P_{\text{log-rank}}$ = 0.049). Compared with endotype A (clinical^classic^microbiome^M. nonliquefaciens^inflammation^IFN-intermediate^) infants, the rate of developing recurrent wheeze by age 3 years was not significantly different in endotype C or D infants. By contrast, the rate was significantly higher in endotype B (clinical^atopic^microbiome^S. pneumoniae/M. catarrhalis^inflammation^IFN-high^) infants (HR 5.50; 95% CI 1.22–24.8; $P$ = 0.03). Corresponding hazards ratio estimates are presented in Table 2. **b** Recurrent wheeze by age 3 years without asthma at age 5 years. The survival curves did not significantly differ across the endotypes ($P_{\text{log-rank}}$ = 0.84). Compared with endotype A infants, the rate of developing recurrent wheeze by age 3 years was not significantly different in endotype B, C, or D. Corresponding hazards ratio estimates are presented in Table 2. RSV respiratory syncytial virus, IFN interferon, IL interleukin.

host transcriptome data, we selected 3000 transcripts with high variances after normalization and variance stabilizing transformation using DESeq2 package[47]. For the metabolome data, we selected 100 metabolites with high variances with adjusting for batch effects using sva package[48,49] after log$_2$ transformation. Then, we computed Gower distance for the clinical data using StatMatch package[50], Bray–Curtis distance for the microbiome data using vegan package[51], Pearson distance for the transcriptome data using amap package[52], and Euclidian distance for the metabolome data using amap package[52].

To conduct integrated omics endotyping, we computed an affinity matrix of each dataset separately, and computed a fused affinity matrix by similarity network fusion[53] using SNFtool package[54]. We set all parameters of similarity network fusion (i.e., the number of neighbors [$n$ = 25], hyperparameter [alpha = 0.7], and number of iteration [$T$ = 25]). Finally, to identify mutually exclusive endotypes, we conducted spectral clustering using fused affinity matrix. To choose the optimal number of endotypes, we chose the optimal number of endotypes by using a combination of the average silhouette scores (Supplementary Fig 2a), network modularity (Supplementary Fig 2b), endotype size ($n$ = 43–63), and clinical and biological plausibility. The network modularity measures how well separated subnetworks are given a particular partitioning (i.e., endotypes) of the network[55]. To complement these approaches, we have also used a priori knowledge. Indeed, these derived endotypes are characterized by major clinical characteristics (e.g., atopy), major airway bacteria (e.g., *S. pneumoniae*, *M. catarrhalis*, *Haemophilus influenzae*), and immune response profiles (e.g., type I interferons): (A) clinical^classic^microbiome^M. nonliquefaciens^inflammation^IFN-intermediate^, (B) clinical^atopic^microbiome^S. pneumoniae/M. catarrhalis^inflammation^IFN-high^, (C) clinical^severe^microbiome^mixed^inflammation^IFN-low^, and (D) clinical^non-atopic^microbiome^M.catarrhalis^inflammation^IL-6^. These endotypes are consistent with earlier studies that individually investigated each of clinical[56,57] and omics (transcriptome[58], microbiome[7,8], and metabolome[32]) data in infants with bronchiolitis. We conducted differential expression gene and functional pathway analyses by comparing the reference endotype with each of the other endotypes. We also computed the ranking of normalized mutual information (NMI) score of each variable. Each variable is ranked based on the similarity in the clustering of fused matrix, meaning that high-ranked variables contribute more to form the similarity network[53]. After deriving these endotypes, we visualized a patient similarity network with Fruchterman-Reingold layout using qgraph package[59].

We also conducted several analyses to examine the between-endotype differences in clinical and virus, microbiome, and metabolome data. First, we examined the between-endotype differences in the patient characteristics and clinical presentation by using Kruskal–Wallis, chi-squared, and Fisher exact tests, as appropriate. Second, to examined the relationship between major clinical variables and endotypes, we developed a chord diagram, Venn diagram of three major clinical variables (parental history of asthma, IgE sensitization, rhinovirus infection) and their intersections, and an upset plot corresponding to the presented

Venn diagram. We used circlize package[60] for the chord diagram, VennDiagram package[61] for the Venn diagram, and ComplexUpset package[62] for the upset plot. Lastly, to visualize the between-endotypes differences in the clinical characteristics, viruses, selected microbiota species, and selected metabolites, we developed a heatmap assigning the mean value for clinical variables and viruses, microbiome, and metabolome data. The clinical variables and viruses are treated as numeric variables and processed by autoscaling. The microbiome data (20 most abundant species) are processed by log$_2$ transformation and autoscaling. The metabolome data (30 highest NMI scores) are processed by autoscaling.

For transcriptome data, we conducted differential gene expression analysis and functional pathway analysis by comparing the reference endotype (endotype A) with each of the other three endotypes. To visualize the differentially expressed genes for each of the comparisons, we used a volcano plot and created a heatmap using 300 genes with the smallest $P$ values. To investigate whether genes for specific biological pathways are enriched among the large positive or negative fold changes, we conducted a functional class scoring analysis using fgsea package[63]. To detect biologically meaningful pathways by integrating host transcriptome and metabolome data, we also performed a Wilcoxon pathway enrichment analysis using the Integrated Molecular Pathway Level Analysis (IMPaLA) method[64].

To determine the association of endotypes with the risk of childhood asthma (binary outcome), we fitted a logistic regression model. To examine the longitudinal relations of endotypes with the rate of recurrent wheeze outcome, we modeled the time to outcome development (i.e., the development of recurrent wheeze) by fitting a Cox proportional hazards model. Patients who did not have an outcome were censored at their last follow-up interview during the 36-month follow-up period. The proportionality of hazards assumption was verified through evaluating the Schoenfeld residuals. To examine the robustness of endotype-outcome associations, we also conducted a series of sensitivity analyses—(1) excluding infants with coinfection with rhinovirus, (2) using all variables from the full datasets, and (3) a different number of endotypes—and examined the between-endotype difference in the clinical, microbiome, transcriptome, and metabolome data for the major between-group comparison. We analyzed the data using R version 3.6.1 (R Foundation for Statistical Computing, Vienna, Austria). All $P$ values were two-tailed, with $P < 0.05$ considered statistically significant. We corrected for multiple testing using the Benjamini–Hochberg FDR method[65].

**Reporting summary**. Further information on research design is available in the Nature Research Reporting Summary linked to this article.

## Data availability

All relevant data that support the findings of this study will be available on the NIH/NIAID ImmPort and/or dbGaP through controlled access or from the authors. To be

compliant with the informed consent forms of the MARC-35 study and the genomic data sharing plan, the data are available only for research that studies the possible genetic causes of severe bronchiolitis, recurrent wheezing, asthma, and related concepts. BBMap used in this study can be found in the following site: https://sourceforge.net/projects/bbmap/files/ The expanded Human Oral Microbiome Database (eHOMD) database used in this study can be found in the following repository: http://www.homd.org. Source data are provided with this paper.

## Code availability

Computational code from the study is available at https://github.com/HasegawaLab/sample_code_snf_nc_open. Yoshihiko Raita, Integrated omics endotyping of infants with respiratory syncytial virus bronchiolitis and risk of childhood asthma, HasegawaLab/sample_code_snf_nc_open: Sample code for similarity network fusion, https://doi.org/10.5281/zenodo.4731225.

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

## Acknowledgements

This study was supported by grants from the National Institutes of Health (Bethesda, MD): U01 AI-087881, R01 AI-114552, R01 AI-108588, R01 AI-134940, and UG3/UH3 OD-023253. M.P.-L. was partially supported by the Margaret Q. Landenberger Research Foundation, the NIH National Center for Advancing Translational Sciences (Award Number UL1TR001876), and the Fundação para a Ciência e a Tegnologia (T495756868-00032862). The content of this manuscript is solely the responsibility of the authors and does not necessarily represent the official views of the National Institutes of Health. The funding organizations were not involved in the collection, management, or analysis of the data; preparation or approval of the manuscript; or decision to submit the manuscript for publication. We thank the MARC-35 study hospitals and research personnel for their ongoing dedication to bronchiolitis and asthma research (Supplementary Table 11), and Ashley F. Sullivan, MS, MPH and Janice A. Espinola, MPH (Massachusetts General Hospital, Boston, MA) for their many contributions to the MARC-35 study. We also thank Alkis Togias, MD, at the National Institutes of Health (Bethesda, MD) for helpful comments about the study results.

## Author contributions

Y.R. carried out the main statistical analysis, drafted the initial manuscript, and approved the final manuscript as submitted. M.P.L. conducted microbiome and transcriptome analyses, carried out statistical analysis, reviewed the manuscript, and approved the final manuscript. R.J.F. and B.H. conducted specimen processing, supervised RNA sequencing and data generation, reviewed and revised the manuscript, and approved the final manuscript as submitted. J.M.M. collected the data, reviewed and revised the manuscript, and approved the final manuscript as submitted. P.A.P. conducted virus testing and interpreted the results, reviewed and revised the manuscript, and approved the final manuscript as submitted. Z.Z. assisted statistical analysis, reviewed the manuscript, and approved the final manuscript. C.A.C. conceptualized and designed the study, obtained funding, collected the data, supervised the conduct of the study and the analysis, critically reviewed and revised the initial manuscript, and approved the final manuscript as submitted. K.H. conceptualized the study, obtained funding, supervised the statistical analysis, reviewed and revised the initial manuscript, and approved the final manuscript as submitted.

## Competing interests

The authors declare no competing interests.
