## [Peer Review File · Nature Communications]

Reviewer comments, first round: –

Reviewer #1 (Remarks to the Author):

In this manuscript, Raita et al use transcriptomics and metabolomics to define four different endotypes of RSV bronchiolitis and characterize the relationship between endotype and asthma risk. Overall, the authors are addressing an important clinical issue via integration of novel multi-omics datasets, and this will be of broad interest to clinicians, microbiologists and researchers interested in multi-omics. Nevertheless, I have several major concerns, as follows:

Major concerns:

1. Rationale for the number of endotypes should be strengthened. Silhouette plot shows extremely minimal differences in silhouette score between endotypes, particularly between 3, 4 and 6 endotypes. Authors should provide a stronger rationale for the selection of four endotypes than these minor differences, particularly given the constant emphasis on the choice of four endotypes throughout the manuscript. "Clinical plausibility" is likewise insufficient, particularly given the lack of detail provided. This is further enhanced by the fact that alluvial analysis provides rationale for alternative endotype numbers, with separation between and within endotypes C and D particularly weak. A stronger rationale could be visible and distinct observation of four separate clusters by hierarchical clustering methods or distance analyses, as long as these are performed on the full dataset, rather than on metabolites or transcripts pre-selected to be differential (see comment below). Alternatively, authors may wish to focus on endotypes A and B, and indicate insufficient support to distinguish between remaining samples or to determine additional endotype segregation.
2. Was the silhouette analysis performed on the fused data? This should be clearly stated. Silhouette analysis on each individual omics dataset may be more informative.
3. Performing distance analysis only using the most variable metabolites, transcripts or microbes is inappropriate and artificially maximizes differences between groups. At minimum, authors should also provide distance analyses on the full dataset.
4. Instrumental parameters for metabolomics are insufficient to ensure reproducibility. Authors should refer to the reporting guidelines of the Metabolomics Standards Initiative for guidance (PMC3772505). Specific missing items include all liquid chromatography parameters (gradient, gradient duration, flow rate, column temperature, specific mobile phase) and all mass spectrometer source parameters (e. g. vacuum pressure, voltage, gas flow, source temperature).
5. Metabolite extraction methods are insufficient. The standards used for analysis should be specified, and the ratio of methanol to airway sample should be provided.
6. Figure 3b heatmap data does not reflect the matching data in Supplementary table 3. For example, median values for Dihomolinolenate in endotypes B and D are negative, so they should be in the white to blue color range, but instead they are displayed in the red color range. Why is the color for gulonate in endotype C the reddest of the heatmap, when BHBA in endotype C has an even higher metabolite intensity listed in Supplementary table 3? Please clarify.
7. Please provide statistical support for the statements made in lines 96 to 109. For example, it is essential to confirm that IgE sensitization frequency is higher in endotype B than in all the other groups (not just that IgE sensitization differs between groups overall).
8. Table 1 and Supplementary Table 3: please provide post-hoc test results to demonstrate which endotypes have significant differences in the displayed characteristics.
9. Supplementary figure 3: please provide a post-hoc test to demonstrate the specific endotypes where these microbial species differ
10. Supplemental methods, lines 192-194. Using raw p-values of 0.05-0.1 is excessively lenient. More appropriate would be to use FDR-corrected p-values < 0.1
11. Supplementary figure 3: was any multiple hypothesis correction performed?
12. Supplementary figure 3: only very minor differences between groups are observed. Can the authors comment on the biological relevance of these small changes, and how they relate to method accuracy and reproducibility? What is the purpose of displaying non-significant boxplots?
13. Figure 4a and Supplementary figure 5a, 6a and 9a: is the displayed p-value and cutoff on the

volcano plot FDR-corrected? If not, it should be.

14. Line 131, line 133, line 138, are p values FDR-corrected? If not, they should be and this should be specified.

15. What is the source of the IFN and IL-6 data? Is it from the transcriptomic analyses? I could not find that data in the figures or the tables.

16. Data availability statement and main manuscript line 389: "The data that support the findings of this study shall be available on the NIH/NIAID ImmPort through controlled access to be compliant with the informed consent forms of MARC-35 study". "Shall be available" is insufficient. Please ensure that data is available.

Minor comments:

1. I question the utility of the integrated microbiome and transcriptome enrichment analysis, given that many of the returned pathways are as vague as "Metabolism" or "Disease".

2. Supplementary table 2: I am surprised that birth weight showed statistically significant differences between analytic and non-analytic cohorts, given that the same median and IQR are reported. Can the authors verify and confirm?

3. Reporting summary, Materials & experimental systems: why is "clinical data" set to N/A?

4. Supplementary methods, line 126: Ensure that m/z is italicized.

5. Main manuscript, line 390: why is "compliant" italicized? Same comment for "with", line 207.

6. Lines 250, 251: italics emphasis is unnecessary.

7. Figure 1 panel a typo: "Spectoral clustering" should read "Spectral clustering"

Reviewer #2 (Remarks to the Author):

In this study, Raita and colleagues integrated clinical, viral, airway microbiome, transcriptome and metabolome data from samples from a prospective multicenter cohort of 221 children hospitalized with RSV bronchiolitis. By integrating these different pieces of information, they managed to identify 4 endotypes, one of which is distinguished by a significantly higher risk of developing asthma.

To my knowledge, this is one of the very first studies to determine a classification of RSV bronchiolitis into different endotypes, simultaneously integrating clinical, biological and microbiological data - and to analyze these endotypes in relation to the risk of chronic morbidities, such as asthma.

The results presented in this study open the door to much larger prospective studies and could provide clinicians with valuable evidence for the early identification of high-risk children. The data obtained is also an important starting point for the future development of prophylactic and therapeutic strategies for the treatment of bronchiolitis and asthma.

General comments

This study is really very interesting, and the results brought open very important perspectives in this field of research. The work carried out is considerable, bearing in mind that the determination of the microbiome, transcriptome and metabolome from nasopharyngeal samples represents a very important technical challenge.

My major criticism of this paper is not related to the results or their interpretation, but mainly to their representation. The integrated analysis carried out in this study is quite complex and would deserve a little more descriptive and explanatory elements in the main manuscript. From my point of view, this essential part of the study cannot only be present in the supplementary data. Clearly, the most interesting part of the manuscript is mainly found in the supplementary data... In addition, although the current figures are of high quality, I think that the figures could be organised differently and provided with more complete legends.

My other criticisms are essentially points raised by the authors themselves in the discussion. A major limitation of this work is that it is based on the analysis of upper respiratory samples, even though the pathology is located in the lower tract, whose microbiological, transcriptomic and

metabolomic characteristics are known to be quite different. Another major point is the size of the sample, given the number of variables taken into account in the study, and despite a large cohort of 221 patients.

My most technical criticism is that the analysis of differential gene expression as well as the functional analysis was carried out by comparing one endotype to another, and not against a common reference, which ideally would have been a non-infected control.

Specific comments

Introduction - Compared to the discussion, which is particularly well-constructed, the introduction of the manuscript is particularly short, probably a bit succinct to allow readers to better understand the issues and objectives of the study.

Results section - It would have been interesting to have a very short description of MARC-35 at the beginning of the section devoted to the results, in parallel with the exhaustive description given as additional information, in order to have a more precise context of the constitution of the cohort.

Results section - The list of the principal investigators in the 17 sites participating in MARC-35, and could appear at the end of the supplementary data (instead of the beginning) to facilitate the analysis of the (numerous) supplementary data.

At the beginning of the results section, the authors state that they have identified 4 distinct endotypes among children with RSV bronchiolitis. All of the steps and methodology used are not described, and in order to find this information readers should explore and pick up informations in the materials and methods as well as the supplementary data. Despite the constraints of the guidelines, in terms of manuscript size, it seems nevertheless essential to describe this part in more detail in the manuscript.

Results section - Concerning coinfections, in addition to the 72% of simple RSV infections and the 13% of Rhinovirus coinfections, there are still 15% of undescribed coinfections. This is a significant percentage (more than 30 patients) higher than the RSV/Rhinovirus coinfections - do we have any idea what kind of coinfections are involved in this case? It seems important to have an idea of which pathogens are involved.

Results section - The whole part concerning integrated omics endotyping is not sufficiently described, in particular the descriptive information concerning matrix calculations and clustering. The rest of the manuscript is based on the 4 endotypes and therefore it seems indispensable to describe how they were determined. More globally, all the comparative analysis of endotypes is presented in supp figures, and almost nothing is described nor explained in the main manuscript.

What is the purpose of Fig. 2? A more precise description in the legend seems necessary to enable the reader to better understand what the network really represents. In this type of representation, there seems to be a significant overlap between endotypes, especially between endotypes A and D. Note: the colour code of the knots is not exactly the same as the one shown in the legend. This figure could have been completed with results originally shown in the supp figure (especially the supp Figs 2 and 3).

Fig.1 part a "Network"

Dear the Reviewers:

We appreciate the time spent by you and believe the revised manuscript is improved. Below, we have addressed your comments.

Sincerely,

Yoshihiko Raita, MD, MPH, MMSc on behalf of the authors.
Department of Emergency Medicine, Massachusetts General Hospital, Harvard Medical School,
Boston, MA, USA (on behalf of all the authors)

COMMENTS FROM REVIEWER #1:

In this manuscript, Raita et al use transcriptomics and metabolomics to define four different endotypes of RSV bronchiolitis and characterize the relationship between endotype and asthma risk. Overall, the authors are addressing an important clinical issue via integration of novel multi-omics datasets, and this will be of broad interest to clinicians, microbiologists and researchers interested in multi-omics. Nevertheless, I have several major concerns, as follows:

[Response]

We thank you for these positive comments.

MAJOR COMMENTS:

1. Rationale for the number of endotypes should be strengthened. Silhouette plot shows extremely minimal differences in silhouette score between endotypes, particularly between 3, 4 and 6 endotypes. Authors should provide a stronger rationale for the selection of four endotypes than these minor differences, particularly given the constant emphasis on the choice of four endotypes throughout the manuscript. “Clinical plausibility” is likewise insufficient, particularly given the lack of detail provided. This is further enhanced by the fact that alluvial analysis provides rationale for alternative endotype numbers, with separation between and within endotypes C and D particularly weak. A stronger rationale could be visible and distinct observation of four separate clusters by hierarchical clustering methods or distance analyses, as long as these are performed on the full dataset, rather than on metabolites or transcripts pre-selected to be differential (see comment below). Alternatively, authors may wish to focus on endotypes A and B, and indicate insufficient support to distinguish between remaining samples or to determine additional endotype segregation.

[Response]

We thank you for this thoughtful comment. As suggested, in the revised manuscript, we have chosen the optimal number of endotypes by using a combination of the average silhouette scores (**Supplementary Fig 2, Panel a**), network modularity (**Supplementary Fig 2, Panel b**),

endotype size (n=43-63), and clinical and biological plausibility. We have added the network modularity since we have applied network approaches. The network modularity measures how well separated subnetworks are given a particular partitioning (i.e., endotypes) of the network¹. We have confirmed that the network modularity was highest with $k=4$ (**Supplementary Fig 2, Panel b**; see the plot below). To complement these approaches, we have also used *a priori* knowledge. Indeed, these derived endotypes are characterized by major clinical characteristics (e.g., atopy), major airway bacteria (e.g., *S. pneumoniae*, *M. catarrhalis*, *Haemophilus influenzae*), and immune response profiles (e.g., type I interferons): A) clinical^{classic} microbiome^{*M. nonliquefaciens*} inflammation^{IFN-intermediate}, B) clinical^{atopic} microbiome^{*S. pneumoniae/M. catarrhalis*} inflammation^{IFN-high}, C) clinical^{severe} microbiome^{mixed} inflammation^{IFN-low}, and D) clinical^{non-atopic} microbiome^{*M. catarrhalis*} inflammation^{IL-6}. These endotypes are consistent with earlier studies that individually investigated each of clinical^{2,3} and omics (transcriptome⁴, microbiome^{5,6}, and metabolome⁷) data in infants with bronchiolitis. Moreover, we acknowledge that there is *no* universal standard to choose *the* number of endotypes. Therefore, we have also performed a sensitivity analysis with $k=5$. As presented in **Supplementary Table 7**, there is a strong concordance, particularly in the endotypes A and B (i.e., our focus), between $k=4$ and $k=5$. In addition, the associations between the endotypes and asthma did not change materially with the use of $k=5$ (**Supplementary Table 8**). These consistencies lend additional support to our inference.

As suggested, we have highlighted these points in the Methods section (lines, 522-540). The text now states: “To choose the optimal number of endotypes, we chose the optimal number of endotypes by using a combination of the average silhouette scores (**Supplementary Fig 2, Panel a**), network modularity (**Supplementary Fig 2, Panel b**), endotype size (n=43-63), and clinical and biological plausibility. The network modularity measures how well separated subnetworks are given a particular partitioning (i.e., endotypes) of the network¹. To complement these approaches, we have also used *a priori* knowledge. Indeed, these derived endotypes are characterized by major clinical characteristics (e.g., atopy), major airway bacteria (e.g., *S. pneumoniae*, *M. catarrhalis*, *Haemophilus influenzae*), and immune response profiles (e.g., type I interferons): A) clinical^{classic} microbiome^{*M. nonliquefaciens*} inflammation^{IFN-intermediate}, B) clinical^{atopic} microbiome^{*S. pneumoniae/M. catarrhalis*} inflammation^{IFN-high}, C) clinical^{severe} microbiome^{mixed} inflammation^{IFN-low}, and D) clinical^{non-atopic} microbiome^{*M. catarrhalis*} inflammation^{IL-6}. These endotypes are consistent with earlier studies that individually investigated each of clinical^{2,3} and omics (transcriptome⁴, microbiome^{5,6}, and metabolome⁷) data in infants with bronchiolitis.” Furthermore, as suggested, the manuscript has focused mainly on the endotypes A and B (e.g., the Results section lines 183-202; **Fig. 5**).

Supplementary Fig. 2. Average silhouette score and network modularity, according to number of endotypes

a. Average silhouette score, according to number of endotypes

Across the different numbers of endotypes (k of 2-6), the average silhouette score was highest with $k=4$.

b. Network modularity, according to number of endotypes

Across the different numbers of endotypes (k of 2-6), the network modularity was highest with $k=4$.

2. Was the silhouette analysis performed on the fused data? This should be clearly stated. Silhouette analysis on each individual omics dataset may be more informative.

[Response]

We appreciate the opportunity to clarify the point. We have computed the average silhouette scores on the fused affinity. We note that the objective of the study is not to *individually* examine the role of each omics (e.g., microbiome, transcriptome, and metabolome, as previously published) but to examine their *integrated* role of these multiple omics data. Accordingly, we have computed a fused affinity matrix and applied spectral clustering to this fused affinity matrix. Nevertheless, as suggested, we have performed an exploratory analysis to compute average silhouette scores on each dataset (see the plot below). The clinical and virus data have the largest average silhouette scores among the four datasets. In the clinical and virus data (and in the metabolome data), the average score is highest with $k=4$, which is consistent with those computed from the fused matrix.

As suggested, we have also clarified this point in the **RESULTS** section. The text now states that “Lastly, to identify mutually-exclusive endotypes, we applied spectral clustering to the fused affinity matrix. We used a combination of the average silhouette scores, network modularity, endotype size, and clinical and biological plausibility to choose the optimal number of endotypes. Across the different numbers of endotypes (k of 2-6), both the average silhouette score and the network modularity were highest with $k=4$ (**Supplementary Fig. 2**).” (lines 117-121).

a Clinical and virus data

b Microbiome data

c Transcriptome data

c Metabolome data

3. Performing distance analysis only using the most variable metabolites, transcripts or microbes is inappropriate and artificially maximizes differences between groups. At minimum, authors should also provide distance analyses on the full dataset.

[Response]

We appreciate the opportunity to clarify this point. We agree with you that an examination of multi-omics data with a high dimensionality (e.g., 19,056 transcripts) imposes an important challenge. This is a balancing act between variable selection and potential biases from the noise with the use of all data. Nonetheless, we note that our aims are 1) to identify biologically distinct RSV bronchiolitis endotypes (i.e., description [clustering]) and 2) to relate them to the outcome risks (i.e., association)". Accordingly, for the first goal, we do not make inference on endotyping using ground truth labels. Rather, endotyping is a data-driven, hypothesis-generating approach that facilitates further investigations. Variable selection methods (e.g., selecting variables with high variances) have traditionally been applied to derive phenotypes and endotypes⁸⁻¹¹.

Regardless, as suggested, we have repeated the analysis using *all* variables from the full datasets. An additional alluvial plot (**Supplementary Fig. 7**; see page 8 of this letter) shows a consistency between the original endotypes (A-D) and endotypes using all variables from the full datasets (1-4). For example, the endotype 1 has an 88.4% concordance with the original endotype A and the endotype 2 has an 82.5% concordance with the original endotype B (see the table next page). The Rand index that represents the concordance between two clustering results is 0.87 (95%CI, 0.83-0.90). In agreement with the original analysis, compared to endotype 1 infants, endotype 2 infants had a significant higher risk of asthma (14.6% vs. 36.4%; OR, 3.35; 95%CI, 1.35–9.19; P=0.012; **Supplementary Table 6**; see next page).

The text now states that “we repeated the analyses without excluding the pre-processed variables. Alluvial plot (**Supplementary Fig. 7**) demonstrated consistency between the 4 endotypes derived in this sensitivity analysis and the original 4 endotypes (A-D). Compared to infants with an endotype 1 (which corresponds to endotype A), those with an endotype 2 (which corresponds to endotype B) had a significant higher risk of asthma (14.6% vs. 36.4%; OR, 3.35; 95%CI, 1.35–9.19; P=0.012; **Supplementary Table 6**).” (lines 213-218).

Concordance Table.

	Endotype 1 (n=48)	Endotype 2 (n=66)	Endotype 3 (n=47)	Endotype 4 (n=60)
Original 4 endotypes (A-D)				
Endotype A (n=43): clinical ^{classic} microbiome ^{M. nonliquefaciens} inflammation ^{IFN-intermediate}	38 (88.4%)	0 (0)	2 (4.7%)	3 (7.0%)
Endotype B (n=63): clinical ^{atopic} microbiome ^{S. pneumoniae/M. catarrhalis} inflammation ^{IFN-high}	10 (15.9%)	52 (82.5%)	0 (0)	1 (1.6%)
Endotype C (n=63): clinical ^{severe} microbiome ^{Mixed} inflammation ^{IFN-low}	0 (0)	14 (22.2%)	45 (71.4%)	4 (6.3%)
Endotype D (n=52): clinical ^{non-atopic} microbiome ^{M. catarrhalis} inflammation ^{IL-6}	0 (0)	0 (0)	0 (0)	52 (100%)

Supplementary Table 6. Association of respiratory syncytial virus bronchiolitis endotypes using all variables in infants with development of asthma and recurrent wheeze outcomes

Endotypes	Childhood asthma at age 5 years*		Recurrent wheeze by age 3 years [†] with asthma [‡]		Recurrent wheeze by age 3 years [†] without asthma [§]	
	Odds ratio (95% CI)	P-value	Hazard ratio (95% CI)	P-value	Hazard ratio (95% CI)	P-value
Endotype 1	1 [Reference]	-	1 [Reference]	-	1 [Reference]	-
Endotype 2	3.35 (1.35–9.19)	0.012	2.09 (0.70–6.24)	0.19	2.24 (0.90–5.55)	0.082
Endotype 3	1.20 (0.40–3.73)	0.74	0.97 (0.26–3.63)	0.97	1.61 (0.61–4.23)	0.33
Endotype 4	1.46 (0.54–4.26)	0.46	1.53 (0.50–4.67)	0.46	1.54 (0.61–3.92)	0.36

Abbreviation: CI, confidential interval

* Asthma (binary outcome) was defined as physician-diagnosis of asthma by age 5 years, plus either asthma medication use (e.g., albuterol inhaler, inhaled corticosteroids, montelukast) or asthma-related symptoms in the preceding year. To examine the association between RSV bronchiolitis endotypes (endotype A as the reference) and the risk of developing childhood asthma, logistic regression model was fit.

[†] Recurrent wheeze (time-to-event outcome) was defined as having at least two corticosteroid-requiring exacerbations in six months or at least four wheezing episodes in one year that last at least one day and affect sleep. To examine the association between RSV bronchiolitis endotypes (endotype A as the reference) and the rate of recurrent wheeze, Cox proportional hazards model was fit.

[‡] The outcome is recurrent wheeze by age 3 years *with* epidemiological definition of asthma (n=26) vs. no recurrent wheeze (n=127). The analysis excludes the other children (n=68).

§ The outcome is recurrent wheeze by age 3 years *without* epidemiological definition of asthma (n =43) vs. no recurrent wheeze (n=127). The analysis excludes the other children (n=51).

Supplementary Fig. 7. Alluvial plot that examines consistencies between the original endotypes and endotypes derived from the sensitivity analysis including all the variables

4. Instrumental parameters for metabolomics are insufficient to ensure reproducibility. Authors should refer to the reporting guidelines of the Metabolomics Standards Initiative for guidance (PMC3772505). Specific missing items include all liquid chromatography parameters (gradient, gradient duration, flow rate, column temperature, specific mobile phase) and all mass spectrometer source parameters (e. g. vacuum pressure, voltage, gas flow, source temperature).

[Response]

As you requested, we have added the information of instrumental parameters used in the metabolome profiling to the **Supplementary Methods** and **Supplementary Table 9** (see next page). The text now states that “Sample preparation was carried out as described previously^{12,13}. In brief, recovery standards were added prior to the first step in the extraction process for quality control purposes. Proteins were precipitated with 500 μ L of methanol added to 100 μ L of sample under vigorous shaking for two minutes (Glen Mills Genogrinder 2000; Clifton, NJ, USA) followed by centrifugation. The sample extract was dried then reconstituted in solvents compatible to each of the four methods. Each reconstitution solvent contained a series of standards at fixed concentrations to ensure injection and chromatographic consistency. One aliquot was analysed using acidic positive ion conditions, chromatographically optimized for hydrophilic compounds. In this method, the extract was gradient eluted from a C18 column (Waters UPLC BEH C18-2.1 \times 100 mm, 1.7 μ m) using water and methanol, containing 0.05% perfluoropentanoic acid and 0.1% formic acid. Another aliquot was also analysed using acidic positive ion conditions, but it was chromatographically optimized for hydrophobic compounds. In this method, the extract was gradient eluted from the same aforementioned C18 column using methanol, acetonitrile, water, 0.05% perfluoropentanoic acid and 0.01% formic acid and was operated at an overall higher organic content. Another aliquot was analysed using basic negative ion optimized conditions using a separate dedicated C18 column. The basic extracts were gradient eluted from the column using methanol and water, including 6.5 mM Ammonium Bicarbonate at pH 8. The fourth aliquot was analysed via negative ionization following elution from a HILIC column (Waters UPLC BEH Amide 2.1 \times 150 mm, 1.7 μ m) using a gradient consisting of water and acetonitrile with 10mM Ammonium Formate, pH 10.8. The MS analysis alternated between MS and data-dependent MSⁿ scans using dynamic exclusion. The scan range varied slightly between methods but covered 70-1000 *m/z*. Parameters for chromatography are summarized in **Supplementary Table 9.**” (**Supplementary Methods**; lines 186-206).

Supplementary Table 9. Chromatography condition for metabolome profiling

Chromatography condition	Positive ionization chromatographically optimized for hydrophilic compounds	Positive ionization chromatographically optimized for hydrophobic compounds	Negative ionization optimized conditions	Negative ionization with HILIC chromatography
Column	Waters UPLC BEH C18-2.1x100 mm, 1.7 μm	Waters UPLC BEH C18-2.1x100 mm, 1.7 μm	Waters UPLC BEH C18-2.1x100 mm, 1.7 μm	Waters UPLC BEH Amide 2.1x150 mm, 1.7 μm
Mobile phase A	0.05% PFPA in water, pH ~2.5 and 0.1% formic acid	0.05% PFPA in water, pH ~2.5 and 0.1% formic acid	6.5 mM ammonium bicarbonate in water, pH 8	10 mM ammonium formate in 15% water/ 5% methanol/ 80% acetonitrile (effective pH 10.16 with NH ₄ OH)
Mobile phase B	0.1% formic acid and 0.05% PFPA in methanol, pH ~2.5	0.1% formic acid and 0.05% PFPA in 50% methanol/ 50% acetonitrile, pH ~2.5	6.5 mM ammonium bicarbonate in 95% methanol/ 5% water	10 mM ammonium formate in 50% water/ 50% acetonitrile (effective pH 10.80 with NH ₄ OH)
Flow rate	0.35 mL/min	0.60 mL/min	0.35 mL/min	0.50 mL/min
Gradient elution	Linear gradient from 5% B to 80% B over 3.35 minutes	Linear gradient from 40 % B to 99.5% B over 1.0 minute, hold 99.5% B for 2.4 minutes.	Linear gradient from 0.5 to 70% B over 4.0 minutes, then rapid gradient to 99% B in 0.5 minutes	Linear gradient from 5% B to 50% B over 3.5 minutes, then linear gradient from 50% B to 95% B in 2 minutes
Spray voltage (V)	4000	4200	3300	3000
Mass range (m/z)	70-1000	110-1000	80-1000	80-1000
Sheath gas (au)	70	35	70	60
Auxiliary gas (au)	35	35	15	20
Source temp (□)	300	400	300	300
Ion transfer tube temp (□)	250	320	250	250
Norm collision energy (au)	52, 65, 78	52, 65, 78	52, 65, 78	48, 60, 72
MS Automatic Gain Control target (au)	1.00E+06	1.00E+06	1.00E+06	1.00E+06
MS Max Fill Time (ms)	60	60	60	60
MS ⁿ Ion Target (au)	2.00E+05	2.00E+05	2.00E+05	2.00E+05

MS ⁿ Max Fill Time (ms)	120	120	120	120
MS ⁿ Isolation Window (m/z)	3	3	3	3
MS ⁿ Dynamic Exclusion Time (s)	3	3	3	3
S-Lens RF Level	40	50	40	25

Abbreviations: MS, mass spectroscopy; HILIC, hydrophilic interaction liquid chromatography; PFPA, perfluoropentanoic acid; UPLC, ultra-high-performance liquid chromatography; *m/z*, mass-to-charge ratio

5. Metabolite extraction methods are insufficient. The standards used for analysis should be specified, and the ratio of methanol to airway sample should be provided.

[Response]

We appreciate the opportunity to clarify this point. As described in the response above, Metabolon (Durham, NC) used the ration of methanol to nasopharyngeal airway sample of 5:1. As suggested, we have added this information to the **Supplementary Methods**. The text now states that “Proteins were precipitated with 500 μ L of methanol added to 100 μ L of sample under vigorous shaking for two minutes (Glen Mills Genogrinder 2000; Clifton, NJ, USA) followed by centrifugation...”(lines 187-189).

6. Figure 3b heatmap data does not reflect the matching data in Supplementary table 3. For example, median values for Dihomolinolenate in endotypes B and D are negative, so they should be in the white to blue color range, but instead they are displayed in the red color range. Why is the color for gulonate in endotype C the reddest of the heatmap, when BHBA in endotype C has an even higher metabolite intensity listed in Supplementary table 3? Please clarify.

[Response]

We appreciate the opportunity to clarify this point. We have presented the metabolite intensities using \log_2 transformation in the original Supplementary Table 3. In the original Fig. 3b heatmap, we used the metabolite intensity values with \log_2 transformation and scaling. After scaling, we cannot compare the intensity between different metabolites. In other words, the goal of the heatmap is to visually present the between-endotype differences in each of the metabolites—i.e., not between the metabolites. Regardless, as suggested, we have revised the heatmap by consistently using autoscaling for the microbiome, metabolome data. We have fixed the colour bars as appropriate, replaced the original one with the figure below (Fig. 2), and revised the figure legend.

Fig. 2. Between-endotype differences in clinical variables, virus, and nasopharyngeal microbiome in infants with respiratory syncytial virus bronchiolitis

To visualize the between-endotype differences, the clinical variables and viruses are treated as numeric variables and processed by auto scaling. The microbiome data (20 most abundant species) are processed by \log_2 transformation and auto scaling. The metabolome data (20 metabolites with highest normalized mutual information score) are processed by \log_2 transformation with batch effect adjustment and autoscaling

7. Please provide statistical support for the statements made in lines 96 to 109. For example, it is essential to confirm that IgE sensitization frequency is higher in endotype B than in all the other groups (not just that IgE sensitization differs between groups overall).

[Response]

We thank you for this comment. We note that the goal of clustering in the current study is to derive RSV bronchiolitis endotypes (i.e., description) (lines 491-492). More generally, the goal of any clustering by itself is not statistical inference but description¹⁴. To be consistent with this goal, we have not made a statistical inference (e.g., post-hoc “hypothesis” testing) for each of many clinical variables, but we have described the endotypes using the clinical data. Accordingly, we have already stated in the manuscript that “Descriptively, infants with an endotype A were characterized by “classic” clinical presentation of bronchiolitis...” (lines 135-147). To help readers understand these differences, we have presented the between-endotype differences in the clinical variables, virus, microbiome, and metabolome using a heatmap (**Fig. 2**; see the previous page) and the relationship between the endotypes and the major clinical variables using chord diagram, Venn diagram, and upset plot (**Fig. 3 a-c**; see below and the next page).

Fig. 3. Relationship between major clinical variables and endotypes

a. Chord diagram showing major clinical variables by endotype

The ribbons connect from the individual endotypes to the major clinical and virus characteristics. The width of ribbon represents the proportion of infants within the endotype who have the corresponding clinical or virus characteristic scaled by the size of endotype. For example, the endotype B infants (light red) had a high proportion of parental asthma, IgE sensitization, and coinfection with rhinovirus. Endotype C (light orange) infants had a high proportion of lifetime antibiotics use and positive pressure ventilation use during the index hospitalization for bronchiolitis.

b. Venn diagram of three major clinical variables (parental history of asthma, IgE sensitization, rhinovirus infection) and their intersections

Venn diagram illustrates the composition of three major clinical variables and their intersections. The numbers correspond to the number of infants in each subset and intersection.

c. Upset plot, corresponding to the presented Venn diagram

The plot illustrates the composition of three major clinical variables and their intersections visualized based on the four endotypes. Vertical stacked bar charts reflect the number of infants within each subset and intersection coloured according to the endotypes. Horizontal bars indicate the number of infants in each clinical variable set. Black dots indicate the sets of subsets and intersections; connecting lines indicate relevant intersections related to each stacked bar chart.

8. Table 1 and Supplementary Table 3: please provide post-hoc test results to demonstrate which endotypes have significant differences in the displayed characteristics.

[Response]

We thank you for this question. As responded to the previous comment, the goal of clustering by itself is *not* statistical inference *but* description¹⁴. In addition, adding a post-hoc test (${}_4C_2 = 6$ post-hoc tests) for each of the clinical variables would not only make Table 1 hard to understand for readers but also lead to a multiple testing problem. Regardless, we have presented the between endotypes A-B comparisons of variables in **Supplementary Table 3**.

9. Supplementary figure 3: please provide a post-hoc test to demonstrate the specific endotypes where these microbial species differ

[Response]

As suggested, we have presented the comparison of microbiome data between the endotypes A and B (the main comparison groups) in **Supplementary Table 3**.

10. Supplemental methods, lines 192-194. Using raw p-values of 0.05-0.1 is excessively lenient. More appropriate would be to use FDR-corrected p-values<0.1

[Response]

We thank you for this comment. As you pointed out, in the over-representation analysis, we need to set an arbitrary threshold. Besides, the over-representation analysis cannot account for 1) gene-gene correlations or 2) ranking and relative scoring of genes (i.e., the importance of the gene at the top of the list is identical to that of the bottom]). In contrast, the gene set enrichment analysis can incorporate gene-level statistics for all genes from the differential expression results. This enables us to investigate if gene sets for particular biological pathways are upregulated or downregulated among the large positive or negative fold changes. Accordingly, we have removed over-representation analyses from the manuscript.

11. Supplementary figure 3: was any multiple hypothesis correction performed?

[Response]

As requested, we have added the raw P-values of Kruskal-Wallis test and the false discovery rates to the Figure (see below). In addition, we have moved this figure to **Fig. 4.** of the main manuscript.

12. Supplementary figure 3: only very minor differences between groups are observed. Can the authors comment on the biological relevance of these small changes, and how they relate to method accuracy and reproducibility? What is the purpose of displaying non-significant boxplots?

[Response]

We appreciate the opportunity to clarify this point. We have presented the boxplot (see the previous page) at the relative abundance scale (range 0-1), not at the absolute abundance scale. Microbes constitute the ecological diversity, and hence the relative species abundance in each endotype measures how common or rare a specific species is relative to other species within each endotype. While the differences in the relative value may appear modest, for example, the difference in the median relative abundance of *S. pneumoniae* is actually large and meaningful (0.05 in the endotype A and 0.26 in B). To avoid any confusion, we have clarified that the Y-axis of **Fig. 4** indicates the relative abundances of each species.

13. Figure 4a and Supplementary figure 5a, 6a and 9a: is the displayed p-value and cutoff on the volcano plot FDR-corrected? If not, it should be.

[Response]

As requested, we have changed values of Y-axis of the volcano plots. The threshold of \log_2 fold change is $|\log_2(1.5)|$ (i.e., $\geq |1.5|$ -fold change) and that of $FDR < 0.1$.

14. Line 131, line 133, line 138, are p values FDR-corrected? If not, they should be and this should be specified.

[Response]

As requested, we have presented raw P-values and false discovery rates. For example, the text now states: “Similarly, the overall host transcriptome profile was different between these endotypes (**Fig. 5A heatmap**) with 29 differentially-expressed genes ($FDR < 0.1$ with $\geq |1.5|$ -fold change; **Fig. 5A volcano plot**; **Supplementary Table 4**).” (lines 187-202).

15. What is the source of the IFN and IL-6 data? Is it from the transcriptomic analyses? I could not find that data in the figures or the tables.

[Response]

We appreciate the opportunity to clarify this point. To investigate whether genes for specific biological pathways are enriched among the large positive- or negative-fold changes, we have conducted the functional pathway analysis using Hallmark pathways (**Fig. 5b, Supplementary Figs. 4b and 5b**). For example, compared with endotype A infants, endotype B infants had 24 differentially-enriched pathways ($FDR < 0.05$)—e.g., up-regulated IFN- α and - γ pathways (**Fig. 5b**; please see below). In contrast, the endotype C infants had 11 differentially-enriched pathways ($FDR < 0.05$)—e.g., down-regulated IFN- α and - γ pathways (**Supplementary Fig. 4**). The endotype D infants had an up-regulated IL-6 pathway (**Supplementary Fig. 5**). As suggested, we have summarized these results in the Results section—e.g., “In the functional pathway analysis that identifies biologically-meaningful pathways, endotype B infants had 24 differentially-enriched pathways ($FDR < 0.05$)—e.g., up-regulated IFN- α and - γ pathways (Fig. 5B).” (lines 198-200).

Fig. 5. Panel b

b Functional pathway analysis

16. Data availability statement and main manuscript line 389: “The data that support the findings of this study shall be available on the NIH/NIAID ImmPort through controlled access to be compliant with the informed consent forms of MARC-35 study”. “Shall be available” is insufficient. Please ensure that data is available.

[Response]

We thank you for the comment. In addition to the importance of data sharing, the compliance with the informed consent with the study subjects comes first. The informed consent forms of MARC studies states “Your child’s samples and information will be used to study the possible genetic causes of severe bronchiolitis, recurrent wheezing, asthma and related concepts”. Thus, the datasets will be available through controlled access. Our Data Sharing Plan approved by the NIH/NIAID states “Human genetic data will be released no later than six months after the data have been submitted to dbGaP and/or NIAID ImmPort, or at the time of acceptance of the first primary publication, whichever occurs first, without restrictions on publication or other dissemination of research findings since use of data is granted following approval from the NIH to use the requested data for a particular project.”

As requested, we have revised the data availability statement. The text now states that “The data that support the findings of this study will be available on the NIH/NIAID ImmPort and/or dbGaP through controlled access to be compliant with the informed consent forms of MARC-35 study.”

Using these procedures, we have already shared our data with researchers around the world.

Minor comments:

1. I question the utility of the integrated microbiome and transcriptome enrichment

analysis, given that many of the returned pathways are as vague as “Metabolism” or “Disease”.

[Response]

We appreciate the opportunity to clarify this point. The goal of the integrated enrichment analysis¹⁵ using transcriptome and metabolome data is to identify enriched pathways in which both genes and metabolites are linked through complex biological pathways. We agree with you that some of the enriched pathways are generic (e.g., Metabolism). This can happen in most pathway analyses. However, we also note that specific dysregulated pathways have been identified. For example, in the comparison of endotype A with endotype B, the endotype B infants had enriched PI3K-Akt-mTOR-signaling and EGFR1 pathways (all FDR<0.05; **Fig. 5c**). The literature has shown the role of PI3K-Akt-mTOR-signaling in the pathobiology of asthma, including activating both innate and adaptive immunity, and airway remodelling. PI3K inhibition not only reduces allergen-induced inflammation and hyperresponsiveness, but also prevents expression of IFN- γ -induced protein—a mediator released by virus-induced asthma. Likewise, in the comparison of endotype A with other endotypes, several specific enriched pathways are identified (e.g., GPCR downstream signaling, L13a-mediated translational silencing of ceruloplasmin expression, and EGFR1 pathways in the comparison of endotype A of endotype C in **Supplementary Fig. 4**; EGFR1 pathways in the comparison of endotype A of endotype D in **Supplementary Fig. 5**).

2. Supplementary Table 2: I am surprised that birth weight showed statistically significant differences between analytic and non-analytic cohorts, given that the same median and IQR are reported. Can the authors verify and confirm?

[Response]

The medians and interquartile ranges of birth weight were rounded to integer values. To avoid confusion, we have now presented the original values—median 3.20 (IQR 2.85-3.54) kg in the analytic cohort and 3.30 (2.90-3.60) in the non-analytic cohort—in **Supplementary Table 1**.

3. Reporting summary, Materials & experimental systems: why is “clinical data” set to N/A?

[Response]

We have revised the reporting summary.

Reporting for specific materials, systems and methods

We require information from authors about some types of materials, experimental systems and methods used in many studies. Here, indicate whether each material, system or method listed is relevant to your study. If you are not sure if a list item applies to your research, read the appropriate section before selecting a response.

Materials & experimental systems

- n/a | Involved in the study
- Antibodies
 - Eukaryotic cell lines
 - Palaeontology and archaeology
 - Animals and other organisms
 - Human research participants
 - Clinical data
 - Dual use research of concern

Methods

- n/a | Involved in the study
- ChIP-seq
 - Flow cytometry
 - MRI-based neuroimaging

4. Supplementary methods, line 126: Ensure that m/z is italicized.

[Response]

Done.

5. Main manuscript, line 390: why is “compliant” italicized? Same comment for “with”, line 207.

[Response]

Done.

6. Lines 250, 251: italics emphasis is unnecessary.

[Response]

We have removed the emphasis.

7. Figure 1 panel a typo: “Spectral clustering” should read “Spectral clustering”

[Response]

We thank you for the careful review. We have fixed the typo in **Fig. 1**.

COMMENTS FROM REVIEWER #2:

In this study, Raita and colleagues integrated clinical, viral, airway microbiome, transcriptome and metabolome data from samples from a prospective multicenter cohort of 221 children hospitalized with RSV bronchiolitis. By integrating these different pieces of information, they managed to identify 4 endotypes, one of which is distinguished by a significantly higher risk of developing asthma. To my knowledge, this is one of the very first studies to determine a classification of RSV bronchiolitis into different endotypes, simultaneously integrating clinical, biological and microbiological data - and to analyze these endotypes in relation to the risk of chronic morbidities, such as asthma.

[Response]

We thank you for these positive comments.

The results presented in this study open the door to much larger prospective studies and could provide clinicians with valuable evidence for the early identification of high-risk children. The data obtained is also an important starting point for the future development of prophylactic and therapeutic strategies for the treatment of bronchiolitis and asthma.

[Response]

We thank you for these positive comments.

General comments

1. This study is really very interesting, and the results brought open very important perspectives in this field of research. The work carried out is considerable, bearing in mind that the determination of the microbiome, transcriptome and metabolome from nasopharyngeal samples represents a very important technical challenge.

[Response]

We thank you for these positive comments.

2. My major criticism of this paper is not related to the results or their interpretation, but mainly to their representation. The integrated analysis carried out in this study is quite complex and would deserve a little more descriptive and explanatory elements in the main manuscript. From my point of view, this essential part of the study cannot only be present in the supplementary data. Clearly, the most interesting part of the manuscript is mainly found in the supplementary data... In addition, although the current figures are of high quality, I think that the figures could be organised differently and provided with more complete legends.

[Response]

As requested, we have re-organized the figures and tables. For example, the figures in the manuscript now present the main concept and findings (please see the response to Comment #10 for the details):

Fig. 1: Analytical workflow for the complex analyses

Fig. 2: Between-endotype differences in clinical variables, virus, and microbiome

Fig. 3: Relationship between major clinical variables and endotypes

Fig. 4: Between-endotype differences in abundance of major bacterial species

Fig. 5: Differential gene expression analysis and functional pathway analysis in the endotypes A vs. B comparison (i.e., the main comparison groups)

Fig. 6: Kaplan-Meier curves for development of recurrent wheeze by age 3 years
Instead, we have moved less important figures—e.g., the network visualization of endotypes—to the **Supplementary Information**.

Secondly, as detailed in the response to the Specific Comments #5, 6, 8, and 10, we have added more descriptive and explanatory elements to the **INTRODUCTION** and **RESULTS** sections.

Lastly, as suggested, we have expanded the figure legends to help readers understand their concepts (e.g., the legend of **Figs. 1-4**; and **Supplementary Fig 3**).

3. My other criticisms are essentially points raised by the authors themselves in the discussion. A major limitation of this work is that it is based on the analysis of upper respiratory samples, even though the pathology is located in the lower tract, whose microbiological, transcriptomic and metabolomic characteristics are known to be quite different. Another major point is the size of the sample, given the number of variables taken into account in the study, and despite a large cohort of 221 patients.

[Response]

We agree with you that bronchiolitis involves inflammation of the lower airways in addition to the upper airways. Although our study is based on the nasopharyngeal samples, studies have shown that upper airway sampling provides reliable representation of the lung microbiome and transcriptomic profiles. Furthermore, the use of upper airway is unavoidable because bronchoscopy or other methods of lower airway sampling would be too invasive in young infants. We have also acknowledged this point in the Discussion section: “Second, bronchiolitis involves inflammation of the lower airways in addition to the upper airways. While our study is based on nasopharyngeal samples, studies have shown that upper airway sampling provides reliable representation of the lung microbiome¹⁶ and transcriptome¹⁷ profiles. Furthermore, the use of upper airway specimens is preferable as bronchoscopy or other methods of lower airway sampling would be too invasive in young infants.” (lines 349-354).

With regard to the sample size (n=221), we also agree with you that an examination of multi-omics data in some study populations (e.g., infants with severe RSV bronchiolitis) imposes an important challenge—e.g., a smaller sample size relative to the number of variables examined, which requires both rigorous statistical approaches (e.g., network and clustering approaches that are used in the current study). We also note that an external validation is needed to validate the novel and biologically-plausible endotypes (while it was already challenging to enroll >1,000 infants with severe RSV bronchiolitis even in this multiyear multicentre cohort study). As suggested, we have highlighted this point in the text, “Considering the relatively small sample size of infants with severe RSV bronchiolitis in the present study, we an external validation is warranted.” (lines 379-380). We have also acknowledged that “Sixth, the sample size of the present analysis is relatively smaller to the number of variables examined. This study should facilitate further validation research.” (lines 362-364).

4. My most technical criticism is that the analysis of differential gene expression as well as the functional analysis was carried out by comparing one endotype to another, and not against a common reference, which ideally would have been a non-infected control.

[Response]

We appreciate the opportunity to clarify this important point. As you pointed out, the current study did not have healthy “controls”. However, the primary objective of this study is not to evaluate the difference between bronchiolitis infants and healthy infants, but to define endotypes within RSV bronchiolitis. Nevertheless, as suggested, we have also acknowledged this point in the Discussion section: “the current study did not have healthy “controls”.” (lines 360-364).

Specific comments

5. Introduction - Compared to the discussion, which is particularly well-constructed, the introduction of the manuscript is particularly short, probably a bit succinct to allow readers to better understand the issues and objectives of the study.

[Response]

As requested, we have expanded the Introduction section. For example, “Understanding the complex interplay among the host, respiratory viruses, airway microbiome, and subsequent chronic morbidities of bronchiolitis involves several major challenges—e.g., identification of the responsible mechanisms (e.g., host-microbiome interrelations), effect of clinical factors reflecting multi-level environmental relations, and heterogeneity of bronchiolitis itself.” (lines 58-62), and “To address this knowledge gap in the literature, analysing data from a multicentre prospective cohort, we sought to 1) identify biologically-distinct RSV bronchiolitis endotypes through applying integrative network and clustering approaches to clinical, virus, nasopharyngeal airway microbiome, transcriptome, and metabolome data and to 2) investigate the association of the derived endotypes with chronic morbidity outcomes (recurrent wheeze by age 3 years and asthma at age 5 years) (**Fig. 1**). We report that four biologically-distinct and clinically-meaningful endotypes are identified: A) clinical^{classic} microbiome^{*M. nonliquefaciens*} inflammation^{IFN-intermediate}, B) clinical^{atopic} microbiome^{*S. pneumoniae/M. catarrhalis*} inflammation^{IFN-high}, C) clinical^{severe} microbiome^{mixed} inflammation^{IFN-low}, and D) clinical^{non-atopic} microbiome^{*M. catarrhalis*} inflammation^{IL-6}. Specifically, ...” (lines 76-85).

6. Results section - It would have been interesting to have a very short description of MARC-35 at the beginning of the section devoted to the results, in parallel with the exhaustive description given as additional information, in order to have a more precise context of the constitution of the cohort.

[Response]

As suggested, we have expanded the Results section. The text now states that “We analysed data from a multicentre prospective cohort study of infants hospitalized for bronchiolitis—the 35th Multicentre Airway Research Collaboration (MARC-35) study. This prospective cohort study completed enrollment of 1016 infants (age <1 year) hospitalized with bronchiolitis at 17 sites across 14 U.S. states. Of these 1,016 infants (median age, 3 months; female, 40%), 921 (91%) completed the run-in procedure (contact at both 1-week after hospital discharge or 3-weeks after hospitalization) and comprise the MARC-35 longitudinal cohort. Of the infants enrolled into this longitudinal cohort, the current study included 221 infants with RSV bronchiolitis who were

randomly-selected for nasopharyngeal microbiome, transcriptome, and metabolome testing (**Supplementary Fig. 1**).” (lines 92-100).

7. Results section - The list of the principal investigators in the 17 sites participating in MARC-35, and could appear at the end of the supplementary data (instead of the beginning) to facilitate the analysis of the (numerous) supplementary data.

[Response]

As suggested, we now present the table as the **Supplemental Table 10**.

8. At the beginning of the results section, the authors state that they have identified 4 distinct endotypes among children with RSV bronchiolitis. All of the steps and methodology used are not described, and in order to find this information readers should explore and pick up informations in the materials and methods as well as the supplementary data. Despite the constraints of the guidelines, in terms of manuscript size, it seems nevertheless essential to describe this part in more detail in the manuscript.

[Response]

As pointed out, we followed the format of *Nature Communications*, in which the Results section (and the Discussion sections) precedes the Methods section that presents more-detailed information on the methods (in addition to the Supplementary Methods).

Regardless, as requested, we have summarized the methods and added it to the Results section. The text now states that “To derive clinically- and biologically-distinct RSV endotypes, we applied integrative network and clustering approaches to clinical, virus, nasopharyngeal airway microbiome, transcriptome, and metabolome data (**Fig. 1**). First, we computed a distance matrix of each dataset—1) Gower distance for clinical and virus data (age, sex, birth weight, history of breathing problems, lifetime antibiotic use, parental asthma, IgE sensitization, positive pressure ventilation use, and virus data), 2) Bray-Curtis distance for microbiome data of 40 most abundant species, which accounted for 95% of total abundance, 3) Pearson distance for transcriptome data using 3,000 gene transcripts with high variances, 4) Euclidian distance for metabolome data with using 100 metabolites with high variances with adjustment for batch effects. Then, we computed an affinity matrix of each dataset separately, and derived a fused affinity matrix by a similarity network fusion. Lastly, to identify mutually-exclusive endotypes, we applied spectral clustering to the fused affinity matrix. To choose the optimal number of endotypes, we used a combination of the average silhouette scores, network modularity, endotype size, and clinical and biological plausibility. Across the different numbers of endotypes (k of 2-6), both of the average silhouette score and the network modularity were highest with $k=4$ (**Supplementary Fig. 2**).” (lines 107-121).

9. Results section - Concerning coinfections, in addition to the 72% of simple RSV infections and the 13% of Rhinovirus coinfections, there are still 15% of undescribed coinfections. This is a significant percentage (more than 30 patients) higher than the RSV/Rhinovirus coinfections - do we have any idea what kind of coinfections are involved in this case? It seems important to have an idea of which pathogens are involved.

[Response]

We appreciate the opportunity to clarify this point. Infants with coinfection by non-rhinovirus include adenovirus infection (n=7), bocavirus (n=8), endemic coronavirus (n=15), enterovirus (n=1), influenza virus (n=1), human metapneumovirus (n=4), *Mycoplasma pneumonia* (n=1), and parainfluenza virus (n=3). Since 6 infants have co-infection with multiple infecting agents, the total number is not equal to 34. As suggested, we have added this information to **Table 1** footnote (see below).

Table 1. Baseline characteristics and clinical course of infants, according to respiratory syncytial virus bronchiolitis endotypes

Characteristics	Overall (n=221; 100%)	Endotype A (n=43; 19.5%)	Endotype B (n=63; 28.5%)	Endotype C (n=63; 28.5%)	Endotype D (n=52; 23.5%)	P-value
Demographics						
Age (month), median (IQR)	3 (2–6)	2 (1–4)	4 (2–6)	4 (2–8)	3 (1–4)	0.002
Female sex	92 (41.6)	35 (81.4)	30 (47.6)	20 (31.7)	7 (13.5)	<0.001
Race/ethnicity						0.40
Non-Hispanic white	92 (41.6)	21 (48.8)	17 (27.0)	32 (50.8)	22 (42.3)	
Non-Hispanic black	54 (24.4)	9 (20.9)	20 (31.7)	12 (19.0)	13 (25.0)	
Hispanic	66 (29.9)	12 (27.9)	22 (34.9)	17 (27.0)	15 (28.8)	
Other or unknown	9 (4.1)	1 (2.3)	4 (6.3)	2 (3.2)	2 (3.8)	
Prematurity (32-37 weeks)	46 (20.8)	7 (16.3)	14 (22.2)	10 (15.9)	15 (28.8)	0.33
Birth weight (kg), median (IQR)	3.20 (2.85–3.54)	3.14 (2.92–3.42)	3.02 (2.81-3.40)	3.30 (2.92–3.58)	3.30 (2.68-3.64)	0.36
Mode of birth (caesarean delivery)	75 (34.6)	13 (30.2)	19 (30.6)	23 (36.5)	20 (40.8)	0.63
Previous breathing problems						0.26
0	189 (85.5)	38 (88.4)	58 (92.1)	48 (76.2)	45 (86.5)	
1	24 (10.9)	3 (7.0)	4 (6.3)	11 (17.5)	6 (11.5)	
2	8 (3.6)	2 (4.7)	1 (1.6)	4 (6.3)	1 (1.9)	
Previous ICU admission	4 (1.8)	0 (0)	0 (0)	3 (4.8)	1 (1.9)	0.21
Lifetime antibiotic use*	67 (30.3)	2 (4.7)	12 (19.0)	48 (76.2)	5 (9.6)	<0.001
Ever attended daycare	66 (29.9)	13 (30.2)	21 (33.3)	18 (28.6)	14 (26.9)	0.89
Cigarette smoke exposure at home	32 (14.5)	11 (25.6)	11 (17.5)	6 (9.5)	4 (7.7)	0.06
Maternal smoking during	30 (13.8)	8 (18.6)	6 (9.7)	10 (15.9)	6 (12.2)	0.56

Parental history of asthma	68 (30.8)	6 (14.0)	50 (79.4)	10 (15.9)	2 (3.8)	<0.001
Parental history of eczema	41 (18.6)	9 (20.9)	14 (22.2)	13 (20.6)	5 (9.6)	0.27
Clinical presentation						
Weight (kg), median (IQR)	5.90 (4.60–7.90)	4.90 (4.14–5.65)	6.20 (5.18–7.75)	6.73 (4.90–8.35)	5.64 (4.40–7.26)	0.003
Respiratory rate (per minute),	48 (40–60)	52 (41–61)	48 (40–62)	45 (38–53)	52 (44–60)	0.045
Oxygen saturation						0.28
<90%	27 (12.6)	5 (11.9)	4 (6.3)	12 (20.3)	6 (11.8)	
90-93%	173 (80.5)	35 (83.3)	55 (87.3)	44 (74.6)	39 (76.5)	
≥94%	15 (7.0)	2 (4.8)	4 (6.3)	3 (5.1)	6 (11.8)	
Blood eosinophilia (≥4%)	18 (9.7)	5 (13.5)	5 (9.8)	3 (5.8)	5 (11.1)	0.64
IgE sensitization	46 (20.8)	7 (16.3)	15 (23.8)	16 (25.4)	8 (15.4)	0.48
Clinical course						
Positive pressure ventilation use [†]	17 (7.7)	2 (4.7)	3 (4.8)	12 (19.0)	0 (0)	0.001
Intensive treatment use [‡]	37 (16.7)	6 (14.0)	10 (15.9)	16 (25.4)	5 (9.6)	0.15
Length-of-day (day), median	2 (1-3)	2 (1–3)	2 (1–4)	2 (1–4)	2 (1–3)	0.17
Antibiotic use during	71 (32.1)	9 (20.9)	21 (33.3)	32 (50.8)	9 (17.3)	<0.001
Corticosteroid use during	24 (10.9)	1 (2.3)	6 (9.5)	13 (20.6)	4 (7.7)	0.02
Respiratory virus						
RSV solo infection	158 (71.5)	32 (74.4)	44 (69.8)	40 (63.5)	42 (80.8)	0.22
Rhinovirus coinfection	29 (13.1)	5 (11.6)	13 (20.6)	5 (7.9)	6 (11.5)	0.21
Rhinovirus-A	14 (6.3)	1 (2.3)	7 (11.1)	5 (7.9)	1 (1.9)	0.15
Rhinovirus-B	4 (1.8)	2 (4.7)	1 (1.6)	0 (0.0)	1 (1.9)	0.37
Rhinovirus-C	11 (5.0)	2 (4.7)	5 (7.9)	0 (0.0)	4 (7.7)	0.09
Other coinfection pathogens [§]	34 (15.4)	6 (14.0)	6 (9.5)	18 (28.6)	4 (7.7)	0.006
Chronic comorbidities						
Asthma at age 5 years	51 (23.1)	4 (9.3)	24 (38.1)	12 (19.0)	11 (21.2)	0.005
Recurrent wheeze by age 3 years	69 (31.2)	10 (23.3)	22 (34.9)	19 (30.2)	18 (34.6)	0.58

Abbreviations: IQR, interquartile range; ICU, intensive care unit; RSV, respiratory syncytial virus; IgE, immunoglobulin E

Data are no. (%) of infants unless otherwise indicated. Percentages may not equal 100, because of rounding and missingness.

* Any systemic antibiotic use from birth up to the index hospitalization for bronchiolitis.

[†] Infants with bronchiolitis who underwent continuous positive airway ventilation and/or mechanical ventilation.

[‡] Infants with bronchiolitis who were admitted to ICU and/or who underwent positive pressure ventilation.

[§]Infants with coinfection by non-rhinovirus include adenovirus infection (n=7), bocavirus (n=8), endemic coronavirus (n=15), enterovirus (n=1), influenza virus (n=1), human metapneumovirus (n=4), *Mycoplasma pneumonia* (n=1), and parainfluenza virus (n=3). Since 6 infants have co-infection with multiple infecting agents, the total number is not equal to 34.

10. Results section - The whole part concerning integrated omics endotyping is not sufficiently described, in particular the descriptive information concerning matrix calculations and clustering. The rest of the manuscript is based on the 4 endotypes and therefore it seems indispensable to describe how they were determined. More globally, all the comparative analysis of endotypes is presented in supp figures, and almost nothing is described nor explained in the main manuscript.

[Response]

As requested, we have expanded the **RESULTS** section in the manuscript. First, we have summarized the descriptive information concerning the matrix calculations and clustering as presented above (please see the response to Comment #8).

Second, as suggested, we have also re-organized the figures and tables. For example, the figures in the manuscript now present the main concept and findings

Figure 1: Analytical workflow for the complex analyses

Figure 2: Between-endotype differences in clinical variables, virus, and microbiome

Figure 3: Relationship between major clinical variables and endotypes

Figure 4: Between-endotype differences in abundance of major bacterial species

Figure 5: Differential gene expression analysis and functional pathway analysis in the endotypes A vs. B comparison (i.e., the main comparison groups)

Figure 6: Kaplan-Meier curves for development of recurrent wheeze by age 3 years

Instead, we have moved less important figures—e.g., the network visualization of endotypes—to the Online Supplement (**Supplementary Fig. 3**).

Third, we note that the **RESULTS** section summarizes the between-endotype differences (particularly between the endotypes A and B—the main comparison groups) in clinical variables, virus, and nasopharyngeal microbiome in infants with respiratory syncytial virus bronchiolitis. For example, the text states that “compared with endotype A infants, endotype B infants had a higher abundance of *S. pneumoniae* but lower abundance of *M. nonliquefaciens* (both $P < 0.05$, $FDR < 0.1$; **Supplementary Table 3**).” (lines 185-187), that “Similarly, the overall host transcriptome profile was different between these endotypes (**Fig. 5a heatmap**) with 29 differentially-expressed genes ($FDR < 0.1$ with $\geq |1.5|$ -fold change; **Fig. 5a volcano plot**; **Supplementary Table 4**). In the gene set enrichment analysis that identifies biologically-meaningful pathways, endotype B infants had 24 differentially-enriched pathways ($FDR < 0.05$)—e.g., up-regulated IFN- α and - γ pathways (**Fig. 5b**).” (lines 185-200), and that “The integrated analysis of both transcriptome and metabolome data also demonstrated 155 differentially-enriched pathways ($FDR < 0.05$)—e.g., upregulated PI3K-Akt-mTOR-signaling pathway in endotype B (**Fig. 4c**). For the endotype A vs. C and endotype A vs. D comparisons, the detailed differences in transcriptome and integrated pathway analysis results are also summarized in **Supplementary Figs. 4-5**.” (lines 202-205).

11. What is the purpose of Fig. 2? A more precise description in the legend seems necessary to enable the reader to better understand what the network really represents. In this type of representation, there seems to be a significant overlap between endotypes, especially between endotypes A and D. Note: the colour code of the knots is not exactly the same as

the one shown in the legend. This figure could have been completed with results originally shown in the supp figure (especially the supp Figs 2 and 3).

[Response]

We appreciate the opportunity to clarify this point. The goal of this figure is a network visualization. We have applied integrative network and clustering approaches to clinical, virus, nasopharyngeal airway microbiome, transcriptome, and metabolome data. This network-based clustering method identifies distinct endotypes based on the degree of connectivity. In other words, infants with similar clinical and biological characteristics are more-closely connected with each other based on the degree of clinical and biological similarity (**Fig. 1**), resulting in the formation of a connected component (i.e., an endotype) in the network. As you suggested, we have now presented this network visualization in the Online Supplement (**Supplementary Fig. 3**; see the next page) and summarized the information above in the figure legend.

We have also fixed the colour Hex code of the legend as appropriate—endotype A, #8DA0CB; endotype B, #E78AC3; endotype C, #FC8D62; endotype D, #66C2A5.

Supplementary Fig. 3. Similarity network visualization of respiratory syncytial virus bronchiolitis endotypes

The goal of this figure is a network visualization. We have applied integrative network and clustering approaches to clinical, virus, nasopharyngeal airway microbiome, transcriptome, and metabolome data. This network-based clustering method identifies distinct endotypes based on the degree of connectivity. In other words, infants with similar clinical and biological characteristics are more-closely connected with each other based on the degree of clinical and biological similarity (**Fig. 1**), resulting in the formation of a connected component (i.e., an endotype) in the network.

Nodes (circles) with a same colour represent infants with a corresponding endotype (A, B, C, or D). Network graphs in the integrated omics similarity network are visualized using Fruchterman-Reingold layout. Layout of the network from the four datasets are fixed with the position of integrated omics similarity network. We selected 1,500 edges with the highest similarity for the network visualization and the width of edge reflects the strength of similarity.

Abbreviations: RSV, respiratory syncytial virus; IFN, interferon; IL, interleukin

REFERENCES

1. Newman, M. E. J. Modularity and community structure in networks. *Proc. Natl. Acad. Sci. U. S. A.* **103**, 8577–8582 (2006).
2. Feldman, A. S., He, Y., Moore, M. L., Hershenson, M. B. & Hartert, T. V. Toward primary prevention of asthma: Reviewing the evidence for early-life respiratory viral infections as modifiable risk factors to prevent childhood asthma. *Am. J. Respir. Crit. Care Med.* **191**, 34–44 (2015).
3. Dumas, O. *et al.* A clustering approach to identify severe bronchiolitis profiles in children. *Thorax* **71**, 712–718 (2016).
4. Fujiogi, M. *et al.* In infants with severe bronchiolitis: dual-transcriptomic profiling of nasopharyngeal microbiome and host response. *Pediatric Research* vol. 88 144–146 (2020).
5. Hasegawa, K. *et al.* Association of nasopharyngeal microbiota profiles with bronchiolitis severity in infants hospitalised for bronchiolitis. *Eur. Respir. J.* **48**, 1329–1339 (2016).
6. Rosas-Salazar, C. *et al.* Differences in the nasopharyngeal microbiome during acute respiratory tract infection with human rhinovirus and respiratory syncytial virus in infancy. *J. Infect. Dis.* **214**, 1924–1928 (2016).
7. Stewart, C. J. *et al.* Associations of nasopharyngeal metabolome and microbiome with severity among infants with bronchiolitis: A multiomic analysis. *Am. J. Respir. Crit. Care Med.* **196**, 882–891 (2017).
8. Ge, S. X., Son, E. W. & Yao, R. iDEP: An integrated web application for differential expression and pathway analysis of RNA-Seq data. *BMC Bioinformatics* **19**, (2018).
9. Xu, T. *et al.* Identifying cancer subtypes from miRNA-TFmRNA regulatory networks and expression data. *PLoS One* **11**, (2016).
10. Pham, V. V. *et al.* Identifying miRNA-mRNA regulatory relationships in breast cancer with invariant causal prediction. *BMC Bioinformatics* **20**, (2019).
11. Vidman, L., Källberg, D. & Rydén, P. Cluster analysis on high dimensional RNA-seq data with applications to cancer research - An evaluation study. *PLoS One* **14**, (2019).
12. Evans, A. M., DeHaven, C. D., Barrett, T., Mitchell, M. & Milgram, E. Integrated, nontargeted ultrahigh performance liquid chromatography/ electrospray ionization tandem mass spectrometry platform for the identification and relative quantification of the small-molecule complement of biological systems. *Anal. Chem.* **81**, 6656–6667 (2009).
13. Ford, L. *et al.* Precision of a clinical metabolomics profiling platform for use in the identification of inborn errors of metabolism. *J. Appl. Lab. Med.* **5**, 342–356 (2020).
14. Hernán, M. A., Hsu, J. & Healy, B. A second chance to get causal inference right: A classification of data science tasks. *CHANCE* **32**, 42–49 (2019).
15. Kamburov, A., Cavill, R., Ebbels, T. M. D., Herwig, R. & Keun, H. C. Integrated pathway-level analysis of transcriptomics and metabolomics data with IMPaLA. *Bioinformatics* **27**, 2917–2918 (2011).
16. Marsh, R. L. *et al.* The microbiota in bronchoalveolar lavage from young children with chronic lung disease includes taxa present in both the oropharynx and nasopharynx. *Microbiome* **4**, (2016).
17. Poole, A. *et al.* Dissecting childhood asthma with nasal transcriptomics distinguishes subphenotypes of disease. *J. Allergy Clin. Immunol.* **133**, (2014).

Reviewer comments, second round: –

Reviewer #1 (Remarks to the Author):

All my comments have been addressed satisfactorily.

Reviewer #2 (Remarks to the Author):

The authors have responded to all comments. I have no other criticisms or suggestions, it is a really interesting work.